# TEMPORAL HETEROGENEOUS GRAPH GENERATION WITH PRIVACY, UTILITY, AND EFFICIENCY

**Xinyu He,**\* **Dongqi Fu**\*, **Hanghang Tong, Ross Maciejewski, Jingrui He**
University of Illinois Urbana-Champaign, Meta AI, Arizona State University
{xhe34, htong, jingrui}@illinois.edu, {dongqifu}@meta.com, {rmacieje}@asu.edu

## ABSTRACT

Nowadays, temporal heterogeneous graphs attract much research and industrial attention for building the next-generation Relational Deep Learning models and applications, due to their informative structures and features. While providing timely and precise services like personalized recommendations and question answering, this rich information also introduces extra exposure risk for each node in the graph. The distinctive local topology, the abundant heterogeneous features, and the time dimension of the graph data are more prone to expose sensitive information and narrow down the scope of victim candidates, which calls for well-defined protection techniques on graphs. To this end, we propose a Temporal Heterogeneous Graph Generator balancing Privacy, Utility, and Efficiency, named **THEPUFF**. More specifically, we first propose a differential privacy algorithm to perturb the input temporal heterogeneous graph for protecting privacy, and then utilize both the perturbed graph and the original one in a generative adversarial setting for THEPUFF to learn and generate privacy-guaranteed and utility-preserved graph data in an efficient manner. We further propose 6 new metrics in the temporal setting to measure heterogeneous graph utility and privacy. Finally, based on temporal heterogeneous graph datasets with up to 1 million nodes and 20 million edges, the experiments show that THEPUFF generates utilizable temporal heterogeneous graphs with privacy protected, compared with state-of-the-art baselines.

## 1 INTRODUCTION

Recently, temporal heterogeneous graphs have emerged as one of the most important fundamental components for the next-generation Relational Deep Learning (Fey et al., 2024), owing to their comprehensive expressiveness. Equipped with temporal heterogeneous graphs, modern Relational Deep Learning models (Robinson et al., 2024) target to provide timely and precise applications like personalized recommendations (Qi et al., 2023; Ban et al., 2024), question answering (Li et al., 2024b), classification and regression (Zheng et al., 2024; Xu et al., 2024a; Wang et al., 2024a), and alignment (Zeng et al., 2023; Yu et al., 2025). During the service-providing process, privacy has become an alarming concern, especially in the era of big data and AI. Then, a natural question arises: *how can we learn the distribution of temporal heterogeneous graphs, such that we can generate utilizable data with privacy-protected and also take the large scalability into account?*

Compared with texts and images, protecting privacy in relational data is even more challenging, because the complex topology introduces additional structural information to expose the uniqueness of entities. For example, in a social network, the neighbors surrounding different users are often different, which can be leveraged by attackers to easily locate the victim and reveal the identity by the isomorphism of the ego-network (i.e., 1-hop neighbors), called neighborhood relationships attacker queries (Zhou & Pei, 2008).

Beyond that, the open-world graphs often involve heterogeneity and temporality (Fu et al., 2022a; Li et al., 2023; Tieu et al., 2024; Lin et al., 2024), which makes the entity's existence even more unique and enlarges the exposure risk. First, a heterogeneous graph means the node type (and edge type) in a graph can be more than one (Shi et al., 2017; Yang et al., 2022). For example, in the citation network, the node types can include 'paper', 'author', and 'venue', and the edge types can have 'cite', 'write', and 'work at'. Second, graph structures (and features) can evolve over time. In a citation network, the structure evolves when new papers are published and new citation relationships are established.

---
\*Equal Contribution

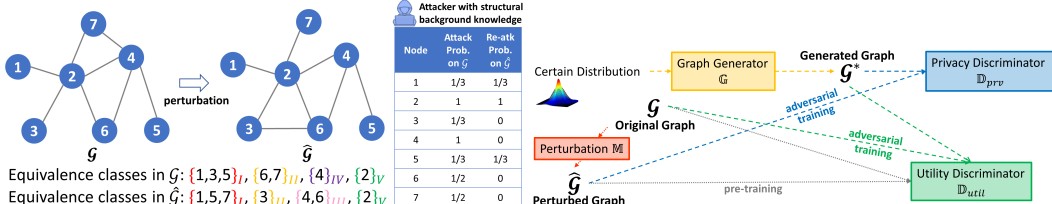

Figure 1: **Left**: A Toy Example of Attack and Protection on a Static Homogeneous Graph: Attacker with Structural Background Knowledge vs. Graph Perturbation based Protection. The second column of the table shows the probability that an attacker can successfully identify the victim with background knowledge, and the third column shows the success probability with the same background knowledge on the perturbed graph. **Right**: Overview of THEPUFF framework.

Overall, the heterogeneous and time-dependent structures and features of original graphs bring extra risk to expose privacy by making nodes more unique, which creates unprecedented challenges for graph generative models to preserve privacy when generating utilizable large-scale graph data.

To bridge this gap, we propose a Temporal Heterogeneous Graph Generator balancing Privacy, Utility, and Efficiency, named THEPUFF. In brief, THEPUFF relies on an adversarial learning manner, where THEPUFF tries to explore the distribution of privacy-guaranteed graphs and utility-preserved graphs and then generates the synthetic graph that preserves both privacy and utility.

To protect privacy and defend attackers, the graph perturbation methods provide a promising direction (Zheleva & Getoor, 2007; Liu & Terzi, 2008; Nguyen et al., 2015; Hoang et al., 2021; Fu et al., 2022b). A motivation case study is shown in Figure 1, where we demonstrate an attacker associated with the background knowledge of *node signature structural query* (Nguyen et al., 2015), i.e., nodes with the same structural information (e.g., node degree) form the victim candidates. The toy example in Figure 1 shows that one simple edge perturbation can decrease this attacker's confidence. For example, the attacker aims to reveal the node identity of node #6 and knows its degree is 2, then in the original graph $\mathcal{G}$, the attacker has 1/2 probability of identifying node #6 correctly. But if graph $\mathcal{G}$ is perturbed into graph $\hat{\mathcal{G}}$, the attacker has 0 probability of identifying node #6, because node #6 is now transferred to another class for the new degree is 3.

Hence, in our THEPUFF, we first propose an error-bounded differential privacy perturbation algorithm for temporal heterogeneous graphs and an efficient and effective privacy-utility adversarial learning method for generating temporal heterogeneous graphs. To demonstrate the performance, we (1) propose 6 metrics for measuring the utility and privacy of temporal heterogeneous graphs to verify the generation, (2) conduct extensive attacking experiments on THEPUFF, (3) design various ablation studies, parameter analysis, and convergence visualization.

## 2 PRELIMINARY ANALYSIS

To make THEPUFF's generation enable privacy-utility balancing and also fit the large-scale graph data, we analyze the current SOTA graph diffusion models (Zhang et al., 2023) and systematically explain (1) why they are insufficient for this setting and (2) why Generative Adversarial Network (Goodfellow et al., 2014) (instanced by Transformers (Vaswani et al., 2017)) is the best fit.

Table 1: Diffusion based Graph Generation

|  | Diffusion type | Complexity | Complexity Bottleneck |
|---|---|---|---|
| DiGress (Vignac et al., 2022) | Denoising Diffusion | $Ln^2$ | link prediction for all possible edges |
| EDP-GNN (Niu et al., 2020) | Score Matching w/ Langevin Dynamics | $Ln^2$ | noise sampling for adjacency entries |
| EDGE (Chen et al., 2023) | Denoising Diffusion | $L(max(k^2, m)), k \sim \frac{1}{10}n$, $m$ is number of edges | link prediction for all possible edges between selected nodes |
| GraphGDP (Huang et al., 2022) | Stochastic Differential Equations | $Ln^2$ | noise sampling for adjacency entries |
| DISCO (Xu et al., 2024b) | Continuous-time Markov Chain | $Ln^2$ | link prediction for all possible edges |

**Diffusion-based Graph Generation.** With the help of diffusion techniques, diffusion graph generation models learn the distribution of the adjacency matrix by diffusing it into an empty or random graph. Despite their success in molecule generation and protein modeling (Wu et al., 2022; Luo et al., 2022), all those methods share a high time complexity on the same scale of $O(Ln^2)$ due to edge-level prediction or noise generation, where $L$ is diffusion steps and $n$ is number of nodes. Previous works are summarized in Table 1. This poses an efficiency concern when dealing with large graphs. Importantly, the diffusion method has inherent advances in privacy protection due to the

noise-adding process, as our experiments show in Sec. 5.2. However, most methods do not provide the privacy guarantee, with the only exception of (Wei et al., 2023) that considers per-instance differential privacy for vector data in discrete diffusion models. Therefore, we choose transformer-based GAN for large-scale and privacy-utility-adversary graph generation, as stated below.

**Transformer-based Generative Adversarial Network (GAN).** With the development of transformers, researchers find that transformers can effectively serve as generators and discriminators of GAN in the context of image generation (Jiang et al., 2021). More recently, transformer-based large language models have also been successfully utilized to be generators and discriminators for fake news detection (Wang et al., 2024b). By representing the graph with extracted random walks, the transformer is also made available for graphs as well (Ling et al., 2021). Moreover, in graph generation, by generating random walks instead of directly generating an adjacency matrix, we can bypass edge-level sampling and prediction, which would lead to high computational costs.

## 3 Notation

We use bold lowercase letters for vectors (e.g., $\boldsymbol{a}$), bold capital letters for matrices (e.g., $\boldsymbol{A}$), and parenthesized superscripts to denote the time indices (e.g., $\boldsymbol{A}^{(t)}$). To be specific, in the graph adjacency matrix, $\boldsymbol{A}(i, j)$ means the connection between node $v_i$ and node $v_j$. Also, we denote the perturbation operation as $\hat{}$, i.e., $\hat{\boldsymbol{A}}$ is the perturbed matrix of $\boldsymbol{A}$, the perturbation operation is related to privacy protection and is introduced in Sec. 4.

**Temporal Heterogeneous Graph Modeling**. Formally, a heterogeneous graph means the number of types of objects (i.e., nodes) is greater than 1 or the number of types of links (i.e., edges) is greater than 1 (Sun et al., 2011). Moreover, the relationship between node type and edge type is deterministic, i.e., if two edges belong to the same type, then the two edges share the same starting node type as well as the ending node type (Sun & Han, 2012). Hence, we use $o$ to denote the node type and do not assign extra symbols for the edge type throughout the paper. For modeling a temporal heterogeneous graph $\mathcal{G}$, we denote each edge as a triplet $((v_i, o_i), (v_j, o_j), t)$, where $o_i$ is the node type of node $v_i$, and $t$ is the timestamp that nodes $v_i$ and $v_j$ connects.

## 4 Proposed Framework: THePUff

As shown in Figure 1, our goal is to generate a synthetic graph $\mathcal{G}^*$ that enjoys both privacy guarantee and high utility. To this end, first, we need a privacy-preserving graph sample $\hat{\mathcal{G}}$ to provide the distribution of privacy-preserved structures. Second, we can then set the original graph $\mathcal{G}$ as the utilizable sample, and develop the adversarial generative model to fit the distribution of $\hat{\mathcal{G}}$ and $\mathcal{G}$ to generate a privacy-preserving and utility-preserving graph $\mathcal{G}^*$. In this section, we first introduce the overall working flow of the proposed THePUff framework in Sec. 4.1. Then, we introduce the details about how to perturb a temporal heterogeneous graph with privacy guaranteed in Sec. 4.2. Finally, we introduce how to design the privacy-utility adversarial learning setting to generate viable temporal heterogeneous graphs in Sec. 4.3.

### 4.1 Overview of THePUff

The overview of THePUff is shown in Figure 1. The red arrows indicate the privacy-guaranteed graph operation, where a perturbation $\mathbb{M}$ is called to transform the original graph $\mathcal{G}$ into $\hat{\mathcal{G}}$, the details are introduced in Sec. 4.2. Then, the blue and green arrows stand for privacy-utility adversarial learning: (1) Privacy Discriminator $\mathbb{D}_{\text{prv}}$ is designed to guarantee the privacy of the generated graph, which aims to discriminate the generated graph $\mathcal{G}^*$ and the perturbed graph $\hat{\mathcal{G}}$; (2) Utility Discriminator $\mathbb{D}_{\text{util}}$ is designed for preserving the graph utility, which aims to discriminate the generated graph $\mathcal{G}^*$ and the original graph $\mathcal{G}$. To optimize this adversarial training,

- We first pre-train $\mathbb{D}_{\text{util}}$ such that it can distinguish an original graph $\mathcal{G}$ with any other graph patterns.
- Then, in the iterations of adversarial training, we alternatively (1) train $\mathbb{D}_{\text{prv}}$ with fixed the Generator $\mathbb{G}$, and (2) train Generator $\mathbb{G}$ with fixed $\mathbb{D}_{\text{prv}}$ and fixed $\mathbb{D}_{\text{util}}$.

After adversarial training, the generation ability of Generator $\mathbb{G}$ is improved, such that (1) Privacy Discriminator $\mathbb{D}_{\text{prv}}$ could not distinguish graph $\mathcal{G}^*$ from $\hat{\mathcal{G}}$ (i.e., privacy guaranteed), and (2) Utility Discriminator $\mathbb{D}_{\text{util}}$ could not distinguish graph $\mathcal{G}^*$ from $\mathcal{G}$ (i.e., utility preserved). Upon this optimization, Generator $\mathbb{G}$ will output the privacy-guaranteed and utility-preserved graph data.

## 4.2 Data-Driven Graph Structure Perturbation based on Differential Privacy

Here, we seek to perturb the input graph $\mathcal{G}$ into $\hat{\mathcal{G}}$, which (1) not only randomizes the graph structural distribution bounded by edge-level differential privacy (2) but also preserves the temporal heterogeneous graph topology in terms of the degree distribution. The main advantage of preserving the topology information includes maintaining the discrimination differently for the discriminator in adversarial training to avoid the training collapse.

Before we introduce the detailed operation, we need to first reformat the temporal heterogeneous graph by a time-respecting representation, i.e., representing all edges (regardless of edge types) that share the same timestamp into one snapshot. This time-respecting representation is more convenient for operation illustration and theoretical analysis and, meanwhile, will not influence follow-up adversarial learning. The graph is represented as $\mathcal{G} = \{\mathcal{G}^{(1)}, \mathcal{G}^{(2)}, \ldots, \mathcal{G}^{(T)}\}$, where each snapshot has the adjacency matrix $\mathbf{A}^{(t)} \in \mathbb{R}^{n \times n}$, and $n$ is the number of all appeared nodes in the lifetime of $\mathcal{G}$ such that $\mathbf{A}^{(t)}$ allows dangling nodes.

**Definition 4.1** (Adjacent Temporal Snapshots). *At the timestamp $t$, two snapshots $\mathcal{G}^{(t)}$ and $\widetilde{\mathcal{G}}^{(t)}$ are adjacent snapshots, if and only if $\mathcal{G}^{(t)}$ and $\widetilde{\mathcal{G}}^{(t)}$ have the same set of nodes and differ in only one edge existence. For example, $e_{ij}$ exists in $\mathcal{G}^{(t)}$ but not in $\widetilde{\mathcal{G}}^{(t)}$, or vice versa.*

More precisely, based on the heterogeneous graph definition (Sun & Han, 2012), the two node types determine the edge type, then the above definition also applies to heterogeneous graphs.

**Definition 4.2** (Adjacent Temporal Graphs). *Two temporal graphs $\mathcal{G}$ and $\widetilde{\mathcal{G}}$ are adjacent, if and only if they have the same time length (i.e., from $t = 1$ to $t = T$) and only one pair of adjacent snapshots exists with the identical rest.*

In the static setting, edge-level differential privacy (Blocki et al., 2012; Qin et al., 2017; Fu et al., 2023) describes that: given two adjacent graphs $G$ and $\widetilde{G}$ (i.e., differ in one edge existence), a randomized mechanism $\mathbb{M}$ satisfies the edge-level differential privacy under a constant budget $\varepsilon > 0$, if and only if

$$Pr[\mathbb{M}(\mathbf{A}) = \mathbf{S}] \leq Pr[\mathbb{M}(\widetilde{\mathbf{A}}) = \mathbf{S}] \times e^{\varepsilon} \tag{1}$$

where $\mathbf{A}$ denotes the adjacency matrix of $G$ and $\widetilde{\mathbf{A}}$ denotes the adjacency matrix of $\widetilde{G}$.

The general idea of the above differential privacy is that two adjacent graphs (e.g., one edge difference between two graphs) are indistinguishable through the perturbation algorithm $\mathbb{M}$. Then, this perturbation algorithm $\mathbb{M}$ satisfies the differential privacy (e.g., edge-level differential privacy). The intuition is that the randomness of $\mathbb{M}$ does not make the small divergence produce a considerably different distribution, such that the randomness of $\mathbb{M}$ is not the cause of the privacy leak. Likewise, the temporal edge-level differential privacy can be defined as follows.

**Definition 4.3** (Temporal Edge-Level Differential Privacy). *Given two adjacent temporal graphs $\mathcal{G} = \{\mathcal{G}^{(1)}, \mathcal{G}^{(2)}, \ldots, \mathcal{G}^{(T)}\}$ and $\widetilde{\mathcal{G}} = \{\widetilde{\mathcal{G}}^{(1)}, \widetilde{\mathcal{G}}^{(2)}, \ldots, \widetilde{\mathcal{G}}^{(T)}\}$, a temporal randomized mechanism $\mathbb{M} = \{\mathbb{M}^{(1)}, \mathbb{M}^{(2)}, \ldots, \mathbb{M}^{(T)}\}$ satisfies the temporal edge-level differential privacy under the budget $\varepsilon > 0$, if and only if*

$$Pr[\mathbb{M}^{(t)}(\mathbf{A}^{(t)}) \in \mathbf{S}^{(t)}] \leq Pr[\mathbb{M}^{(t)}(\widetilde{\mathbf{A}}^{(t)}) \in \mathbf{S}^{(t)}] \times e^{\varepsilon^{(t)}}, and \sum_{t} \varepsilon^{(t)} \leq \varepsilon, \ for \ t \in \{1, 2, \ldots, T\}$$

Next, we introduce our proposed temporal randomized mechanism $\mathbb{M} = \{\mathbb{M}^{(1)}, \mathbb{M}^{(2)}, \ldots, \mathbb{M}^{(T)}\}$, and give the proof under temporal edge-level differential privacy.

**Perturbation Operations**. For a snapshot $\mathcal{G}^{(t)}$ at time $t$, $\mathbb{M}^{(t)}$ contains two independent perturbation operations, i.e., *absent edge flip* $\mathbb{M}_+^{(t)}$ and *existing edge flip* $\mathbb{M}_-^{(t)}$, proposed to be directly applied on adjacency matrix $\mathbf{A}^{(t)}$ to obtain $\hat{\mathbf{A}}^{(t)}$. The *absent edge flip* is responsible for adding an absent edge, and the *existing edge flip* tries to delete an existing edge. These two perturbation operations are sequentially and interchangeably operated on the structure (i.e., adjacency matrix $\mathbf{A}^{(t)}$) of snapshot $\mathcal{G}^{(t)}$ to get the perturbed structure (i.e., adjacency matrix $\hat{\mathbf{A}}^{(t)}$) for the new snapshot $\hat{\mathcal{G}}^{(t)}$. Following this way, we apply $\mathbb{M}^{(t)} = (\mathbb{M}_+^{(t)}, \mathbb{M}_-^{(t)})$ to each snapshot $\mathcal{G}^{(t)}$ such that we can get the perturbed temporal graph $\hat{\mathcal{G}} = \{\hat{\mathcal{G}}^{(1)}, \hat{\mathcal{G}}^{(2)}, \ldots, \hat{\mathcal{G}}^{(T)}\}$. $\mathbb{M}_+^{(t)}$ and $\mathbb{M}_-^{(t)}$ are expressed below.

$$\mathbb{M}_+^{(t)} = \begin{cases} 1/e^{\varepsilon_+^{(t)}} & \text{Prob. of adding an absent edge,} \\ 1 - 1/e^{\varepsilon_+^{(t)}} & \text{Prob. of ignoring that absent edge,} \end{cases} \tag{2}$$

$$\mathbb{M}_{-}^{(t)} = \begin{cases} 1/e^{\varepsilon_{-}^{(t)}} & \text{Prob. of deleting an existing edge,} \\ 1 - 1/e^{\varepsilon_{-}^{(t)}} & \text{Prob. of ignoring that existing edge,} \end{cases} \quad (3)$$

where $e$ is Euler's number, and an absent edge means $\mathbf{A}^{(t)}(i,j) = 0$. Then, given the privacy budget $\varepsilon_{+}^{(t)} > 0$, with the probability $1/e^{\varepsilon_{+}^{(t)}}$, $\mathbb{M}_{+}^{(t)}$ will flip $\mathbf{A}^{(t)}(i,j)$ from 0 to 1; or $\mathbb{M}_{+}^{(t)}$ will do nothing to $\mathbf{A}^{(t)}(i,j)$ with the probability $1 - 1/e^{\varepsilon_{+}^{(t)}}$, similarly on $\varepsilon_{-}^{(t)} > 0$.

When sequentially executing $\mathbb{M}_{+}^{(t)}$ and $\mathbb{M}_{-}^{(t)}$ on a single snapshot graph $\mathcal{G}$, they are executed on different edge sets (i.e., existing edge set and absent edge set) and do not overwrite each other. In practice, given the high sparsity of real-world graphs, iterating through non-existing edges could lead to high time complexity. Therefore, we apply $\mathbb{M}_{+}^{(t)}$ by first sampling a larger edge set (contains mostly non-existing edges and a small portion of existing edges), then remove those existing edges, and sample the number of edges we need to append. In this way, the time complexity of graph perturbation is approximately the number of edges to be sampled. The general graph perturbation process is summarized in Alg. 1 in Appendix A.3.

**Theorem 4.1.** *The proposed perturbation operation for temporal heterogeneous graphs is an $\epsilon$-edge-level differential privacy algorithm, which is bounded by $2 \cdot \max(e^{\varepsilon_{+}^{(t)}}, e^{\varepsilon_{-}^{(t)}})$ at each t. (Proof in Appendix A.1)*

**Data-Driven $\varepsilon^{(t)}$ Determination**. According to our perturbation strategy, the perturbation process may result in a privacy-preserving graph but a totally different topological distribution, thus hurting the utility of the original graph. Also, the dramatic change compared with the original graph is prone to collapse the adversarial training of the next stage.

Hence, we remedy the graph distortion caused by inserted and deleted edges with respect to the degree distribution. To achieve that, from the data-driven angle, we give the relation between $\varepsilon_{+}^{(t)}$ and $\varepsilon_{-}^{(t)}$ for $\mathcal{G}^{(t)}$ based on the expectation of numbers of added edges and detected edges at each time $t$. For the notation clarity, the following equation is expressed in the homogeneous setting but can be easily extended to the heterogeneous graph by constraining node types.

$$m^{(t)} \frac{1}{e^{\varepsilon_{-}^{(t)}}} \approx ((n^{(t)})^2 - m) \frac{1}{e^{\varepsilon_{+}^{(t)}}} \Rightarrow \varepsilon_{-}^{(t)} - \varepsilon_{+}^{(t)} \approx ln \frac{m^{(t)}}{(n^{(t)})^2 - m^{(t)}} \quad (4)$$

where $n^{(t)}$ denotes number of non-dangling nodes in $\mathcal{G}^{(t)}$, $m^{(t)}$ denotes number of edges in $\mathcal{G}^{(t)}$.

### 4.3 Privacy-Utility Adversarial Generation

To integrate the privacy of perturbed graph $\hat{\mathcal{G}}$ and the utility of original graph $\mathcal{G}$, we propose a privacy-utility adversarial learning method. The method consists of three modules: Privacy Discriminator $\mathbb{D}_{\text{prv}}$, Utility Discriminator $\mathbb{D}_{\text{util}}$, and Generator $\mathbb{G}$. During the adversarial training, we extract sampled subgraphs (e.g., via random walks) as model inputs. To be specific, at each time $t$, if a node has connections, we can use a heterogeneous random walk method (Ling et al., 2021) to get multiple $l$-node sequences. The $i$-th sequence is denoted as $\mathcal{W}_i^{(t)} = \{\{v_1^{(t)}, v_2^{(t)}, ..., v_l^{(t)}\}, \{o_1, o_2, ..., o_l\}\}$, where $v_j$ denotes a node and $o_j$ denotes the node type of $v_j$. Finally, we design an assembler for aggregating generated walks of generator $\mathbb{G}$ into a privacy-guaranteed and utility-preserved temporal heterogeneous graph $\mathcal{G}^*$.

**Utility-preserving Discriminator $\mathbb{D}_{\text{util}}$.** This discriminator is designed to distinguish generated graphs $\hat{\mathcal{G}}$ and original graphs $\mathcal{G}$, such that if the generator $\mathbb{G}$ can bypass discriminator $\mathbb{D}_{\text{util}}$, i.e., $\mathbb{D}_{\text{util}}$ cannot distinguish the generated walks sampled from real graphs, then the generated graph is 'close enough' to the real graph and thus preserves the utility. On the other side, given a number of walks sampled from the real graph $\{\mathcal{W}_1, \mathcal{W}_2, ..., \mathcal{W}_k\}$ and some generated walks $\{\mathcal{W}_1^*, \mathcal{W}_2^*, ..., \mathcal{W}_k^*\}$, discriminator $\mathbb{D}_{\text{util}}$ should have a confident bound for

$$\sum_{i=1}^{k} \mathbb{D}_{\text{util}}(\mathcal{W}_i) >> \sum_{i=1}^{k} \mathbb{D}_{\text{util}}(\mathcal{W}_i^*) \quad (5)$$

where we omit superscript for time in $\mathcal{W}_i^{(t)}$, so that $\mathcal{W}_i$ means the $i$-th walk in the sampled walks that consists of walks sampled from different nodes and different timestamps.

To achieve the above goal, we extend our discriminator $\mathbb{D}_{\text{util}}$ from bi-level self-attention (Zhou et al., 2020) to the tri-level self-attention. On top of that, we first use a model-agnostic projection (Hussein

et al., 2018) to map node $v_i^{(t)}$ into an embedding vector $\mathbf{h}_i^{(t)}$. Then, with a sampled walk $\mathcal{W}_i^{(t)} = \{\{v_1^{(t)}, v_2^{(t)}, ..., v_l^{(t)}\}, \{o_1, o_2, ..., o_l\}\}$, tri-level self-attention mechanism $ATN$ is

$$ATN^{time}(v_i^{(t)}, v_j^{(t)}) = \frac{(\mathbf{h}_i^{(t)}\mathbf{W}_Q^{time}) \odot (\mathbf{h}_j^{(t)}\mathbf{W}_K^{time})}{\sqrt{d_k}} \tag{6}$$

$$ATN^{node}(v_i, v_j) = \frac{(\mathbf{h}_i\mathbf{W}_Q^{node}) \odot (\mathbf{h}_j\mathbf{W}_K^{node})}{\sqrt{d_k}}, \quad \mathbf{h}_i = \sum_{t \in \{1, ..., T\}} \mathbf{h}_i^{(t)} \tag{7}$$

$$ATN^{type}(o_i, o_j) = \frac{(\mathbf{o}_i\mathbf{W}_Q^{type}) \odot (\mathbf{o}_j\mathbf{W}_K^{type})}{\sqrt{d_k}} \tag{8}$$

where $\mathbf{o}_i$ is the one-hot embedding of node type $o_i$. Matrices $\mathbf{W}_Q^{time}, \mathbf{W}_Q^{node}, \mathbf{W}_Q^{type}$ are the time-level, node-level, and type-level query weight matrices; $\mathbf{W}_K^{time}, \mathbf{W}_K^{node}, \mathbf{W}_K^{type}$ are the time-level, node-level, and type-level key weight matrices.

To integrate attention from the three levels, we define the pooling operation of tri-level self-attention as
$$\mathbf{Z} = (\text{softmax}(\mathbf{A}^{time}) + \text{softmax}(\mathbf{A}^{node}) + \text{softmax}(\mathbf{A}^{type}))\mathbf{V}, \tag{9}$$

where $\mathbf{V} = \mathbf{W}_V\mathbf{H}$, $H(v, :) = \mathbf{h}_v$, and $\mathbf{W}_V$ is the value weight matrix.

Finally, the multi-head attention layers are followed by a fully connected layer and a softmax function to output a score estimating the probability that the input walk is sampled from the real graph.

To train $\mathbb{D}_{\text{util}}$ for fulfilling the goal in Eq. 5, the objective function for optimizing $\mathbb{D}_{\text{util}}$ is to minimize

$$\mathcal{L}_{\mathbb{D}_{\text{util}}}(\{\mathcal{W}\}, \{\mathcal{W}^*\}) = \sum_i^{|\{\mathcal{W}\}|} \mathbb{D}_{\text{util}}(\mathcal{W}_i^*) - \mathbb{D}_{\text{util}}(\mathcal{W}_i) \tag{10}$$

where $\{\mathcal{W}\}$ and $\{\mathcal{W}^*\}$ are the mini-batches of sampled walks.

For realization, we choose to train $\mathbb{D}_{\text{util}}$ with randomly generated negative samples instead of generated ones from generator $\mathbb{G}$, whose operation actually allows us to detach $\mathbb{D}_{\text{util}}$ from the min-max manner (Goodfellow et al., 2014). In this way, $\mathbb{D}_{\text{util}}$ acts like a good scoring function for the penalty in the adversarial training. Although involving $\mathbb{D}_{\text{util}}$ with the generator might lead to a higher utility, it will increase computation complexity, the difficulty of model tuning, and the risk of training collapse. Our experiments corroborate that training $\mathbb{D}_{\text{util}}$ separately is sufficiently good for generation.

**Privacy-guaranteed Discriminator** $\mathbb{D}_{\text{prv}}$. This discriminator aims to discriminate generated graph $\mathcal{G}^*$ and perturbed graph $\hat{\mathcal{G}}$. Again, if the discriminator $\mathbb{D}_{\text{prv}}$ cannot discriminate walks in the generated graph and walks sampled from the perturbed graph, the privacy of the generated graph can be guaranteed. On the contrary, a good $\mathbb{D}_{\text{prv}}$ should satisfy

$$\sum_{i=1}^k \mathbb{D}_{\text{prv}}(\hat{\mathcal{W}}_i) >> \sum_{i=1}^k \mathbb{D}_{\text{prv}}(\mathcal{W}_i^*) \tag{11}$$

where $\{\hat{\mathcal{W}}_1, \hat{\mathcal{W}}_2, ..., \hat{\mathcal{W}}_k\}$ are random walks sampled from $\hat{\mathcal{G}}$, and $\{\mathcal{W}_1^*, \mathcal{W}_2^*, ..., \mathcal{W}_k^*\}$ are walks generated by generator $\mathbb{G}$.

With similar intuition as $\mathbb{D}_{\text{util}}$, $\mathbb{D}_{\text{prv}}$ shares the same neural architecture as $\mathbb{D}_{\text{util}}$. The objective function for training $\mathbb{D}_{\text{prv}}$ is defined as
$$\mathcal{L}_{\mathbb{D}_{\text{prv}}}(\{\hat{\mathcal{W}}\}, \{\mathcal{W}^*\}) = \sum_i^{|\{\hat{\mathcal{W}}\}|} \mathbb{D}_{\text{prv}}(\mathcal{W}_i^*) - \mathbb{D}_{\text{prv}}(\hat{\mathcal{W}}_i) \tag{12}$$

where $\mathbb{D}_{\text{prv}}$ and generator $\mathbb{G}$ are trained alternatively.

**Generator** $\mathbb{G}$. The generator aims to generate synthetic graphs that are both privacy-guaranteed and utility-preserved. We first introduce how to generate temporal heterogeneous walks, and then we introduce how the generated walks get constrained by the designed loss function to preserve both privacy and utility. In the next subsection, we introduce how the assembler aligns the generated walks into a temporal heterogeneous graph.

To begin with, we sample timestamped nodes with type based on the perturbed graph $\hat{\mathcal{G}}$, which could ensure higher privacy-preserving than just sampling from the original graph $\mathcal{G}$. In the following sections, we omit the time superscript of $v_i^{(t)}$ for brevity, i.e., $v_i$ denotes a node with a timestamp for the following sections. The walk generation relies on the hidden states of recurrent neural architectures

(e.g., LSTM (Hochreiter & Schmidhuber, 1997)) to generate a length-$L$ walk, i.e., the generator sequentially samples timestamped nodes with type based on the hidden state, which corresponding mechanism is described as follows. For sampling the $l$-th node in walk generation, i.e., the $l$-th block in LSTM $f_\theta$, for $l \in \{1, \ldots, L\}$, we have

$$(m_l, h_l) = f_\theta(m_{l-1}, a_{l-1}), \quad a_l = g_c(o_l, v_l),$$
$$o_l \sim g_o(h_l), \quad /* \text{ sample node type } */$$
$$v_l \sim g_v(h_l, o_l), \quad /* \text{ sample timestamped node } */$$

(13)

where $v_l$ is the $l$-th timestamped node, $o_l$ is the node type, $h_l$ and $m_l$ are the hidden state and memory state of LSTM, and $a_l$ is the input for the next recurrent block:

- $g_o$ is implemented with a fully connected layer and Gumbel Softmax, which aims to sample the node type based on the hidden state.

- $g_v$ is implemented by a bag of fully connected layers, each corresponding to one node type; given the sampled node type from $g_o$, the corresponding fully connected layer generates an expected timestamped node embedding $\mathbf{h}^*$. Given that, we can sample from timestamped node embeddings $\hat{\mathbf{h}}^{(t)}$ in $\hat{\mathcal{G}}$ based on the distance of embedding vectors, detailed distance function is stated in Eq. 14.

- $g_c = f_o(\text{onehot}(o_l)) + f_v(\hat{h}_{v_l}^{(t_{v_l})})$, $f_o$ and $f_v$ are both fully-connected layers, where $t_{v_l}$ is the timestamp of node $v_l$.

For initialization, we have $a_0 = 0, m_0 = f_0(z)$, where $f_0$ is a fully connected layer and $z$ is sampled from normal distribution $z \sim \mathcal{N}(0, 1)$. For $g_v$, a timestamped node $v_l$ with type $o_l$ is sampled by the highest probability based on the vector distance as

$$\frac{e^{-||h^* - \hat{h}_{v_l}^{(t_{v_l})}||}}{\sum_{v \text{ s.t. } o(v) = o_l} e^{-||h^* - \hat{h}_v^{(t_v)}||}}$$

(14)

where $o(v)$ means the node type of a node $v$.

Note that we do not enforce adjacent nodes to share the same timestamp to increase the diversity of the generated graph. In the next assembly step, edges connecting nodes from different timestamps will be added to both snapshots.

To generate the privacy-guaranteed and utility-preserving graph, the walks generated by $\mathbb{G}$ try to bypass both $\mathbb{D}_{\text{util}}$ and $\mathbb{D}_{\text{prv}}$. Therefore the loss function for training $\mathbb{G}$ is defined as

$$\mathcal{L}_{\mathbb{G}}(\{\mathcal{W}^*\}) = -\sum_{i=1}^{|\{\mathcal{W}^*\}|} \left( \mathbb{D}_{\text{util}}(\mathcal{W}_i^*) + \mathbb{D}_{\text{prv}}(\mathcal{W}_i^*) \right)$$

(15)

The general training process is summarized in Alg. 4 in Appendix A.3.

**Claim 4.1.** *Since utility discriminator $\mathbb{D}_{util}$ is pre-trained as shown in lines 3–6 of Alg. 2, and the generator $\mathbb{G}$ does not touch any real data as it is only listening to decisions of discriminators, our generative and adversarial game can be rewritten in the form of Eq. 2 and Eq. 3 of (Wu et al., 2019). In (Wu et al., 2019), generative adversarial networks have been proved to have an outstanding generalization property that can be interpreted as constraining the input and output under differential privacy constraints.*

Thus, we only need to make one extra assumption that the sampled temporal heterogeneous subgraphs are independent and identically distributed, and then we can reach a generalization bound of our adversarial learning from the perspective of differential privacy (Wu et al., 2019), which means the adversarial learning is also under differential privacy if the discriminator is well converged (i.e., stable). Note that this assumption is mild, since all subgraphs originate from a whole larger graph, we can assume they are identically distributed; because we independently sample each subgraph, we can also assume they are independently distributed.

**Assembler**. Here, we extend the assembling mechanism in (Ling et al., 2021) to assemble a temporal heterogeneous graph. First, we generate a sufficient number of walks from the generator. Then we construct a scoring matrix $\mathbf{S}$ by counting the existing edges in the generated walks and find all meta-paths in the generated walks. To construct the generated graph, we start with sampling a node $v_1 \sim Pr(v)$ according to probability

$$Pr(v_1 = u) = \frac{\sum_{v \in \hat{\mathcal{G}}}(\mathbf{S}(u, v))}{\sum_{v_i \in \hat{\mathcal{G}}} \sum_{v_j \in \hat{\mathcal{G}}} \mathbf{S}(v_i, v_j)}$$

(16)

where $\mathbf{S}(u, v)$ counts how many connections exist between node $u$ and $v$ in the generated walks.

Then, we sample a meta-path according to the meta-path's frequency. Following that meta-path $\mathbf{O} = (o_1, ...o_l)$, we sequentially sample the next node $v_{i+1}(i \geq 1)$ with node type $o_{i+1}$ according to probabilities proportional to the scoring matrix, where $o(v)$ is the type of node $v$.

$$Pr(v_{i+1} = u) = \frac{\mathbb{1}(o(u) = o_{i+1})\mathbf{S}(v_i, u)}{\sum_{v \in \hat{\mathcal{G}} \wedge o(v) = o_{i+1}} \mathbf{S}(v_i, v)} \tag{17}$$

Finally, we add all edges in the generated meta-path instances into the generated graph. Note that the timestamp of graphs naturally goes with the timestamp of nodes. The overall algorithms and time complexity can be found in Appendix A.3 and A.4.

## 5 EXPERIMENTS

### 5.1 EXPERIMENTAL SETTING

**Datasets**. To test the performance, we utilize 4 real-world publicly-available temporal heterogeneous graph datasets from academic citation graphs (DBLP), online rating graphs (ML-100k, ML-20M), and million-node online shopping graphs (Taobao).

**Baselines**. The comparison baselines include 4 categorical graph generative models. HGEN (Ling et al., 2021) is a heterogeneous graph generation algorithm. TagGen (Zhou et al., 2020) and TG-GAN (Zhang et al., 2021) are temporal graph generation algorithms. EDGE (Chen et al., 2023) is designed for plain graph generation based on diffusion models. DISCO (Xu et al., 2024b) is a discrete-state continuous-time diffusion model for plain graph generation. GraphMaker (Li et al., 2024a) is a diffusion model designed for attributed graph generation. Our THEPUFF stands for the temporal and heterogeneous graph generation. [1]

**Evaluation Metrics**. We propose 6 metrics measuring 4 different aspects of temporal heterogeneous graph generation quality, including plain graph utility, temporal graph utility, heterogeneous graph utility, and overall privacy.

Plain Graph Utility: We propose Node Degree Distribution by Maximum Mean Discrepancy Distance (i.e., MMD), named **Degree** in short, which is expressed as

$$\text{Dist}(\boldsymbol{d}_{ori}, \boldsymbol{d}_{gen}) = \frac{1}{T} \sum_{t=1}^{T} \text{MMD}(\boldsymbol{d}_{ori}^{(t)}, \boldsymbol{d}_{gen}^{(t)}), \tag{18}$$

where $\boldsymbol{d}_{ori}^{(t)}, \boldsymbol{d}_{gen}^{(t)}$ are degrees of nodes in the original and generated graphs at time $t$, and MMD is averaged over all timestamps,

$$\text{MMD}(x, x') = ker(x, x) + ker(x', x') - 2ker(x, x') \tag{19}$$

where the kernel function $ker(\cdot)$ is $\sum_{j=1}^{k} e^{-\alpha_j ||x-x'||^2}$, and $k$ and $\alpha_j$ are constants. In the evaluation, we set $k = 1$ and $\alpha = 4$.

Temporal Graph Utility: We extend the Clustering Coefficient (i.e., **Cluster**), Size of the Largest Connected Component (i.e., **LCC**), and Triangle Count (i.e., **TC**) to the temporal setting, by calculating those metrics at each timestamp and using MMD between the sequences of metrics over time of the original graph $\mathcal{G}$ and the generated graph $\mathcal{G}^*$. Taking LCC as an example,

$$\text{MMD}([\text{LCC}(\mathcal{G}^{(1)}), ..., \text{LCC}(\mathcal{G}^{(T)})], \ [\text{LCC}(\mathcal{G}^{*(1)}), ..., \text{LCC}(\mathcal{G}^{*(T)})]) \tag{20}$$

Heterogeneous Graph Utility: For each timestamp, we count the number of length-2 and length-3 meta-path instances and get a probability distribution of meta-paths. Cross Entropy is applied for length-2 and length-3 respectively to measure distance, and the mean over time is taken to evaluate the overall performance. The corresponding metrics are named **Meta-2** and **Meta-3**. Specifically, we only preserve the meta-paths that exist in the original graph, and all other meta-paths in the generated graph are combined into one category. For example, in the MovieLens 100k dataset, we consider length-2 meta-path distributions over $\mathcal{M}_2 = $ ['user-occupation', 'movie-genre', 'user-movie', 'none of above'],

$$Dist(Pr_{\mathcal{G}}(\mathcal{M}), Pr_{\mathcal{G}^*}(\mathcal{M})) = \frac{1}{T} \sum_{t=1}^{T} \sum_{m \in \mathcal{M}} (-\log \frac{\exp(Pr_{\mathcal{G}^*}^{(t)}(m))}{\sum_{m \in \mathcal{M}} \exp(Pr_{\mathcal{G}^*}^{(t)}(m))} Pr_{\mathcal{G}}^{(t)}(m)) \tag{21}$$

where $Pr_{\mathcal{G}}(m)$ denotes the probability distribution of the meta-path $m$ in the graph $\mathcal{G}$, and $\mathcal{M}$ is a collection of meta-path $m$.

---

[1]Dataset statistics and more implementation details are summarized in Appendix A.5. Code is at https://github.com/xinyuu-he/THePUff.

Overall Privacy: **EO-Rate** is defined as the ratio of overlapped edges (i.e., in terms of existence and edge type) between the original graphs and the generated graph, which is used to analyze the privacy-preserving property of the generation, and is averaged over time.

$$\text{EO-Rate}(\mathcal{G}^*) = \frac{1}{T} \sum_{t=1}^{T} \frac{|edges(\mathcal{G}^{(t)}) \bigcap edges(\mathcal{G}^{*(t)})|}{|edges(\mathcal{G}^{*(t)})|} \tag{22}$$

## 5.2 MAIN RESULTS

In Table 2, we report the average and standard deviation of the generated temporal heterogeneous graphs with respect to the 6 proposed metrics. In general, our proposed THEPUFF framework generates very competitively utilizable temporal heterogeneous graphs compared with the state-of-the-art baselines, especially in the large-scale datasets, DBLP, MovieLens 20M (i.e., ML-20M), and Taobao datasets. Also, we observe that the plain graph generative model, EDGE (Chen et al., 2023) can achieve good performance on small-scale datasets, but it relies on diffusion model for the generation process, which requires costly computational resources, especially high space complexity. We also discuss the convergence of adversarial training in Appendix A.6, briefly analyze the time complexity in Appendix A.4, and show ablation studies in Appendix A.7.

Table 2: Comprehensive Evaluation of Generated Temporal Heterogeneous Graphs, where $e$ denotes scientific notation (e.g., $1e^{-2} = 0.01$), purple, teal, and cyan denote the first, second, and third place

| Datasets | Methods | Utility | | | | | | Privacy |
|---|---|---|---|---|---|---|---|---|
| | | Cluster ($\downarrow$) | TC ($\downarrow$) | LCC ($\downarrow$) | Degree ($\downarrow$) | Meta-2 ($\downarrow$) | Meta-3 ($\downarrow$) | EO-Rate ($\downarrow$) |
| ML-100k | HGEN | $5.871e^{-2}_{(\pm 6.971e^{-4})}$ | $1.000_{(\pm 0.0)}$ | $1.905e^{-1}_{(\pm 0.0)}$ | $1.395e^{-1}_{(\pm 1.353e^{-3})}$ | $1.469_{(\pm 1.850e^{-4})}$ | $1.289_{(\pm 4.450e^{-5})}$ | $3.200e^{-2}_{(\pm 2.008e^{-4})}$ |
| | TagGen | $3.021e^{-1}_{(\pm 1.561e^{-1})}$ | $1.000_{(\pm 0.0)}$ | $6.417e^{-1}_{(\pm 0.023)}$ | $1.570e^{-1}_{(\pm 5.345e^{-3})}$ | $1.500_{(\pm 1.306e^{-1})}$ | $1.310_{(\pm 3.281e^{-3})}$ | $2.238e^{-1}_{(\pm 1.159e^{-1})}$ |
| | TG-GAN | $4.987e^{-1}_{(\pm 7.305e^{-2})}$ | $1.000_{(\pm 0.0)}$ | $2.143e^{-1}_{(\pm 0.024)}$ | $2.097e^{-1}_{(\pm 8.056e^{-2})}$ | $1.527_{(\pm 2.535e^{-2})}$ | $1.308_{(\pm 1.600e^{-5})}$ | $2.666e^{-2}_{(\pm 4.729e^{-3})}$ |
| | EDGE | $3.960e^{-3}_{(\pm 5.544e^{-6})}$ | $1.000_{(\pm 0.0)}$ | $2.955e^{-1}_{(\pm 0.166)}$ | $1.569e^{-1}_{(\pm 4.883e^{-4})}$ | $1.744_{(\pm 6.850e^{-6})}$ | $1.313_{(\pm 0.0)}$ | $5.488e^{-3}_{(\pm 1.143e^{-4})}$ |
| | DISCO | $3.897e^{-1}_{(\pm 3.444e^{-3})}$ | $1.000_{(\pm 0.0)}$ | $1.190_{(\pm 0.0)}$ | $3.632e^{-1}_{(\pm 8.596e^{-3})}$ | $1.653_{(\pm 2.970e^{-3})}$ | $1.313_{(\pm 3.772e^{-5})}$ | $1.854e^{-2}_{(\pm 1.942e^{-4})}$ |
| | GraphMaker | $1.034e^{-3}_{(\pm 4.803e^{-4})}$ | $1.032_{(\pm 0.027)}$ | $1.908e^{-1}_{(\pm 0.001)}$ | $1.385e^{-1}_{(\pm 3.375e^{-3})}$ | $1.490_{(\pm 4.223e^{-3})}$ | $1.302_{(\pm 3.812e^{-4})}$ | $4.330e^{-2}_{(\pm 3.837e^{-3})}$ |
| | THEPUFF | $2.536e^{-3}_{(\pm 5.673e^{-4})}$ | $1.532_{(\pm 0.0)}$ | $2.636e^{-1}_{(\pm 0.071)}$ | $3.547e^{-1}_{(\pm 1.283e^{-2})}$ | $1.664_{(\pm 1.922e^{-3})}$ | $1.313_{(\pm 1.950e^{-5})}$ | $2.247e^{-2}_{(\pm 5.652e^{-3})}$ |
| DBLP | HGEN | $1.088e^{-8}_{(\pm 4.790e^{-10})}$ | $4.270e^{-1}_{(\pm 0.032)}$ | $0.000_{(\pm 0.0)}$ | $1.191e^{-1}_{(\pm 3.784e^{-5})}$ | $1.566_{(\pm 5.550e^{-5})}$ | $0.912_{(\pm 3.059e^{-4})}$ | $3.064e^{-4}_{(\pm 1.623e^{-5})}$ |
| | TagGen | $1.574e^{-2}_{(\pm 1.437e^{-3})}$ | $1.000_{(\pm 0.0)}$ | $0.000_{(\pm 0.0)}$ | $3.339e^{-1}_{(\pm 1.455e^{-1})}$ | $1.821_{(\pm 2.987e^{-2})}$ | $0.919_{(\pm 8.229e^{-3})}$ | $1.164e^{-1}_{(\pm 2.001e^{-2})}$ |
| | TG-GAN | $6.980e^{-3}_{(\pm 6.585e^{-3})}$ | $1.000_{(\pm 0.0)}$ | $0.000_{(\pm 0.0)}$ | $1.496e^{-1}_{(\pm 4.812e^{-2})}$ | $1.794_{(\pm 7.134e^{-3})}$ | $0.916_{(\pm 1.056e^{-2})}$ | $2.090e^{-3}_{(\pm 7.383e^{-4})}$ |
| | EDGE | | | - - - Generates Empty Graph - - - | | | | |
| | DISCO | | | - - - OOM - - - | | | | |
| | GraphMaker | | | - - - OOM - - - | | | | |
| | THEPUFF | $0.000_{(\pm 0.0)}$ | $1.192e^{-7}_{(\pm 0.0)}$ | $2.466e^{-2}_{(\pm 0.023)}$ | $1.265e^{-1}_{(\pm 5.729e^{-2})}$ | $1.839_{(\pm 1.179e^{-4})}$ | $0.915_{(\pm 4.435e^{-4})}$ | $6.908e^{-5}_{(\pm 3.402e^{-5})}$ |
| ML-20M | HGEN | $2.217e^{-8}_{(\pm 4.491e^{-9})}$ | $2.835e^{-1}_{(\pm 0.008)}$ | $2.552e^{-10}_{(\pm 2e^{-10})}$ | $7.843e^{-2}_{(\pm 8.729e^{-3})}$ | $1.091_{(\pm 1.862e^{-4})}$ | $1.232_{(\pm 1.530e^{-4})}$ | $4.266e^{-3}_{(\pm 8.808e^{-5})}$ |
| | TagGen | $7.934e^{-5}_{(\pm 0.0)}$ | $1.008_{(\pm 0.0)}$ | $0.000_{(\pm 0.0)}$ | $1.263e^{-1}_{(\pm 0.0)}$ | $1.532_{(\pm 3.200e^{-6})}$ | $1.266_{(\pm 0.0)}$ | $5.105e^{-1}_{(\pm 0.0)}$ |
| | TG-GAN | $2.154e^{-9}_{(\pm 2.154e^{-9})}$ | $3.056e^{-2}_{(\pm 0.031)}$ | $2.215e^{-1}_{(\pm 0.212)}$ | $5.431e^{-1}_{(\pm 9.484e^{-3})}$ | $1.328_{(\pm 6.003e^{-3})}$ | $1.257_{(\pm 6.818e^{-3})}$ | $2.339e^{-3}_{(\pm 2.915e^{-4})}$ |
| | EDGE | | | - - - OOM - - - | | | | |
| | DISCO | | | - - - OOM - - - | | | | |
| | GraphMaker | | | - - - OOM - - - | | | | |
| | THEPUFF | $0.000_{(\pm 0.0)}$ | $0.000_{(\pm 0.0)}$ | $5.168e^{-1}_{(\pm 0.165)}$ | $5.596e^{-1}_{(\pm 1.225e^{-2})}$ | $1.081_{(\pm 4.282e^{-3})}$ | $1.266_{(\pm 1.000e^{-6})}$ | $2.305e^{-3}_{(\pm 1.856e^{-4})}$ |
| Taobao | HGEN | $0.000_{(\pm 0.0)}$ | $1.192e^{-7}_{(\pm 0.0)}$ | $0.000_{(\pm 0.0)}$ | $6.137e^{-4}_{(\pm 1.382e^{-5})}$ | $1.233_{(\pm 2.706e^{-2})}$ | $0.613_{(\pm 0.0)}$ | $1.127e^{-4}_{(\pm 7.581e^{-6})}$ |
| | TagGen | $0.000_{(\pm 0.0)}$ | $1.192e^{-7}_{(\pm 0.0)}$ | $0.000_{(\pm 0.0)}$ | $4.515e^{-4}_{(\pm 4.168e^{-5})}$ | $1.540_{(\pm 2.941e^{-3})}$ | $0.613_{(\pm 0.0)}$ | $2.082e^{-2}_{(\pm 1.365e^{-4})}$ |
| | TG-GAN | $0.000_{(\pm 0.0)}$ | $1.192e^{-7}_{(\pm 0.0)}$ | $4.982e^{-1}_{(\pm 0.044)}$ | $7.204e^{-4}_{(\pm 5.501e^{-6})}$ | $1.526_{(\pm 2.498e^{-4})}$ | $0.613_{(\pm 0.0)}$ | $5.021e^{-6}_{(\pm 2.063e^{-6})}$ |
| | EDGE | | | - - - OOM - - - | | | | |
| | DISCO | | | - - - OOM - - - | | | | |
| | GraphMaker | | | - - - OOM - - - | | | | |
| | THEPUFF | $0.000_{(\pm 0.0)}$ | $1.192e^{-7}_{(\pm 0.0)}$ | $5.637e^{-1}_{(\pm 0.087)}$ | $7.545e^{-4}_{(\pm 6.616e^{-7})}$ | $1.493_{(\pm 1.116e^{-3})}$ | $0.613_{(\pm 5e^{-7})}$ | $4.367e^{-6}_{(\pm 4.367e^{-6})}$ |

## 5.3 ATTACKER EXPERIMENT

To further evaluate the privacy of our generated graphs, we perform an attack experiment on the DBLP dataset in addition to the EO-Rate metric. We consider the same attack scenario as the toy example in Figure. 1, i.e., at time $t$, nodes sharing the same degree with the target victim are possible candidates. Mathematically, the successful attack probability (i.e., for accurately identifying the target) can be modeled as follows. At a certain timestamp $t$, the attack probability of node $i$ is

$$p_i^{(t)} = \begin{cases} 0 & , \boldsymbol{d}_{gen}^{(t)}[i] \neq \boldsymbol{d}_{ori}^{(t)}[i], \\ \frac{1}{\sum_{v \in \mathcal{G}^*} \mathbb{1}[\boldsymbol{d}_{gen}^{(t)}[v] == \boldsymbol{d}_{gen}^{(t)}[i]]} & , \boldsymbol{d}_{gen}^{(t)}[i] = \boldsymbol{d}_{ori}^{(t)}[i], \end{cases} \tag{23}$$

where $\mathbb{1}$ is indicator function, $\boldsymbol{d}_{gen}, \boldsymbol{d}_{ori}$ are node degree functions of the generated graph and original graph.

Finally, we measure the model's privacy performance by the mean of $p_i^{(t)}$ over timestamps and all nodes. We compare the attack probabilities of graphs generated by THEPUFF, HGEN,

Table 3: Graph Attack on DBLP

| Baseline | Successful Attack Probability (%) |
|---|---|
| Original | 0.093 |
| HGEN | $0.009_{(1.3e^{-4})}$ |
| DPGGAN | $0.008_{(1.3e^{-4})}$ |
| THEPUFF(Ours) | $\mathbf{0.006}_{(9.0e^{-5})}$ |

and a state-of-the-art differential-privacy-based graph generation method, DPGGAN (Yang et al., 2020), and get the following results; we also calculate the successful attack probability on the original graph for reference. The results are shown in Table 3. We can observe that THEPUFF-generated graphs give attackers the lowest confidence of a successful attack, and THEPUFF is also the most stable one with the smallest standard deviation (i.e. numbers inside the parenthesis in Table 3).

## 5.4 PARAMETER ANALYSIS

We analyze the model performance with different choices of $\varepsilon_+$ and $\varepsilon_-$ on MovieLens 100K. THEPUFF is trained with $\varepsilon_- = \{1, 2, 3, 4, 5, 6, 7, 8\}$ and $\varepsilon_+$ is estimated by Eq. 4. Also, the pair of $\varepsilon_-$ and $\varepsilon_+$ is shared across all timestamps $\{1, 2, \ldots, T\}$. We draw the curve of EO-Rate with respect to $\varepsilon_-$ in Figure 2. In general, the smaller budget can preserve more privacy, which aligns with the theoretical differential privacy. EO-Rate increases with $\varepsilon_-$ when $\varepsilon_-$ is larger than 3, because the number of perturbed edges decreases when $\varepsilon_-$ increases, then the perturbed graph will be very close to the real graph. In this way, the synthetic graph will be closer to the real graph too, and its edges will be more likely to overlap with the

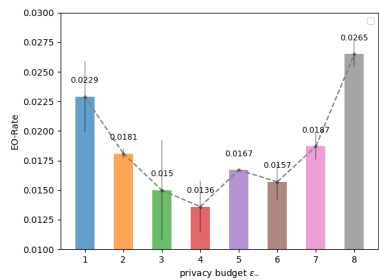

Figure 2: EO-Rate w.r.t Budget $\varepsilon_-$

edges in the real graph. It is interesting that the EO-Rate will also increase when the number of perturbed edges increases beyond a certain boundary ($\varepsilon_-$ less than 3 in our case). This is because the more random the perturbed graph is, the harder the privacy-guaranteed Discriminator's learning process will be. In this case, the utility-preserving discriminator will dominate over the privacy-guaranteed discriminator and generate a synthetic graph more like a real graph. The full results of parameter analysis in terms of all metrics are shown in Table 4.

Table 4: Parameter Study Results on ML-100k, where $e$ denotes the scientific notation.

| $\varepsilon_-$ | Utility | | | | | | Privacy |
|---|---|---|---|---|---|---|---|
| | Cluster ($\downarrow$) | TC ($\downarrow$) | LCC ($\downarrow$) | Degree ($\downarrow$) | Meta-2 ($\downarrow$) | Meta-3 ($\downarrow$) | EO-Rate ($\downarrow$) |
| 1 | $7.598e^{-5}_{(\pm 5.200e^{-5})}$ | $2.000_{(\pm 0.0)}$ | $0.363_{(\pm 2.512e^{-2})}$ | $0.796_{(\pm 0.078)}$ | $1.689_{(\pm 1.824e^{-3})}$ | $1.313_{(\pm 7.300e^{-6})}$ | $2.285e^{-2}_{(\pm 2.968e^{-3})}$ |
| 2 | $1.343e^{-3}_{(\pm 1.166e^{-3})}$ | $1.532_{(\pm 0.0)}$ | $0.240_{(\pm 4.361e^{-4})}$ | $0.413_{(\pm 0.120)}$ | $1.661_{(\pm 2.519e^{-3})}$ | $1.313_{(\pm 4.765e^{-5})}$ | $1.812e^{-2}_{(\pm 4.117e^{-4})}$ |
| 3 | $5.070e^{-5}_{(\pm 3.694e^{-5})}$ | $2.000_{(\pm 0.0)}$ | $0.266_{(\pm 7.448e^{-2})}$ | $0.677_{(\pm 0.117)}$ | $1.676_{(\pm 2.143e^{-3})}$ | $1.313_{(\pm 1.420e^{-5})}$ | $1.500e^{-2}_{(\pm 4.250e^{-3})}$ |
| 4 | $2.540e^{-4}_{(\pm 2.304e^{-4})}$ | $2.000_{(\pm 0.0)}$ | $0.266_{(\pm 7.317e^{-2})}$ | $0.644_{(\pm 0.099)}$ | $1.682_{(\pm 1.773e^{-3})}$ | $1.313_{(\pm 5.970e^{-5})}$ | $1.367e^{-2}_{(\pm 2.199e^{-3})}$ |
| 5 | $3.204e^{-4}_{(\pm 3.054e^{-4})}$ | $2.000_{(\pm 0.0)}$ | $0.240_{(\pm 4.936e^{-2})}$ | $0.603_{(\pm 0.113)}$ | $1.668_{(\pm 5.389e^{-3})}$ | $1.313_{(\pm 1.475e^{-4})}$ | $1.665e^{-2}_{(\pm 8.489e^{-5})}$ |
| 6 | $3.104e^{-4}_{(\pm 2.936e^{-4})}$ | $2.000_{(\pm 0.0)}$ | $0.289_{(\pm 9.611e^{-2})}$ | $0.557_{(\pm 0.119)}$ | $1.669_{(\pm 5.191e^{-3})}$ | $1.313_{(\pm 4.545e^{-5})}$ | $1.575e^{-2}_{(\pm 1.543e^{-3})}$ |
| 7 | $6.915e^{-5}_{(\pm 6.404e^{-5})}$ | $2.000_{(\pm 0.0)}$ | $0.239_{(\pm 1.308e^{-3})}$ | $0.570_{(\pm 0.114)}$ | $1.664_{(\pm 8.025e^{-3})}$ | $1.313_{(\pm 1.002e^{-4})}$ | $1.872e^{-2}_{(\pm 1.095e^{-3})}$ |
| 8 | $3.546e^{-4}_{(\pm 3.204e^{-4})}$ | $1.766_{(\pm 0.234)}$ | $0.265_{(\pm 2.642e^{-2})}$ | $0.489_{(\pm 0.137)}$ | $1.663_{(\pm 1.680e^{-3})}$ | $1.313_{(\pm 5.870e^{-5})}$ | $2.660e^{-2}_{(\pm 1.148e^{-3})}$ |

## 6 RELATED WORK

Graph generative models have been extensively studied recently. Recently, a generative model called EDGE (Chen et al., 2023) is proposed, which develops a diffusion-based model that first diffuses graphs to empty graphs by the step-by-step edge removal process. By modeling that trajectory with the denoising model, it reversely decomposes the denoising model and generates edges. These models are only designed for static and plain graphs. HGEN (Ling et al., 2021) tackles the heterogeneous graph generation problem with a heterogeneous walk-based GAN framework, where a discriminator is trained to discriminate synthetic heterogeneous walks and real walks. TagGen (Zhou et al., 2020) generates synthetic temporal walks by doing 'addition' and 'deletion' to sampled walks. Generated walks are scored to assemble a synthetic temporal graph with discrete timestamps. TG-GAN (Zhang et al., 2021) uses the temporal random walk-based generator-discriminator framework to solve the continuous-time temporal graph generation problem. Most graph generative models ignore the fact that complex heterogeneous and temporal information may increase the exposure of the private information of nodes (e.g., identities of social network users) (Jiang et al., 2023; Fu et al., 2023; Li et al., 2015; Liu et al., 2021), and this challenge largely remains open. To the best of our knowledge, THEPUFF is first to deal with temporal heterogeneous graph generation with guaranteed privacy.

## 7 CONCLUSION

In THEPUFF, a differential privacy algorithm is first proposed to perturb the input graph. Then, a privacy-utility adversarial learning setting is developed to generate both privacy-guaranteed and utility-preserved heterogeneous temporal graphs. By extensive experiments, we demonstrate the effectiveness of THEPUFF, especially in large-scale datasets.

ETHIC STATEMENT

To the best of the authors' knowledge, this research is not related to any ethical issues; all the datasets used in this research are publicly available, and the necessary references and links are provided along with datasets in the main body of the paper.

ACKNOWLEDGMENTS

This work is supported by NSF (2117902, 2324770), and DHS (17STQAC00001-07-00, 17STQAC00001-08-00). The content of the information in this document does not necessarily reflect the position or the policy of the Government, and no official endorsement should be inferred. The U.S. Government is authorized to reproduce and distribute reprints for Government purposes notwithstanding any copyright notation here on.

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

# A APPENDIX

The supplementary material contains the following information:

- Appendix A.1, A.2: Proof of Theorem 4.1.
- Appendix A.3: Pseudo codes.
- Appendix A.4: Time complexity analysis.
- Appendix A.5: Datasets and implementation details.
- Appendix A.6: Convergence of THEPUFF training.
- Appendix A.7: Ablation study.
- Appendix A.8: Attack Experiments on all Datasets
- Appendix A.9: Type-Distinguishable Attack Experiments on all Datasets
- Appendix A.10: Pseudo-code of Discriminators in THEPUFF
- Appendix A.11: Pseudo-code of Assembler in THEPUFF

## A.1 PROOF OF THEOREM 4.1

*Proof sketch.* For the proposed perturbation operation, we first prove that edge-level differential privacy is guaranteed for each timestamp, and we then prove that temporal edge-level differential privacy is guaranteed for the entire temporal graph. Denoting the perturbed adjacency matrix as $\mathbb{M}^{(t)}(\mathbf{A}^{(t)})$, the probability of the perturbed matrix equals a certain matrix $\mathbf{S}^{(t)}$ is

$$Pr[\mathbb{M}^{(t)}(\mathbf{A}^{(t)}) = \mathbf{S}^{(t)}] = \prod Pr[\mathbb{M}_k^{(t)}(\mathbf{A}^{(t)}) = \mathbf{S}_k^{(t)}] \tag{24}$$

where the $\mathbb{M}_k^{(t)}$ includes two randomizations $\mathbb{M}_+^{(t)}$ and $\mathbb{M}_-^{(t)}$.

Then, given two snapshots $\mathcal{G}^{(t)}$ and $\widetilde{\mathcal{G}}^{(t)}$, the ratio of the probability of perturbing on $\mathbf{A}^{(t)}$ and $\widetilde{\mathbf{A}}^{(t)}$ and outputting the same output can be expressed as follows,

$$\frac{Pr[\mathbb{M}^{(t)}(\mathbf{A}^{(t)}) = \mathbf{S}^{(t)}]}{Pr[\mathbb{M}^{(t)}(\widetilde{\mathbf{A}}^{(t)}) = \mathbf{S}^{(t)}]} \text{ s.t. for } i,j \in \{1,2,\ldots,n\} = \prod_{v_i,v_j} \frac{Pr[\mathbb{M}_+^{(t)}(A(i,j)) = S(i,j)]}{Pr[\mathbb{M}_+^{(t)}(\widetilde{A}(i,j)) = S(i,j)]} \prod_{v_i,v_j} \frac{Pr[\mathbb{M}_-^{(t)}(A(i,j)) = S(i,j)]}{Pr[\mathbb{M}_-^{(t)}(\widetilde{A}(i,j) = S(i,j)]} \tag{25}$$

which is a general formula over all possible edges. To be specific, given two adjacent snapshots $\mathcal{G}^{(t)}$ and $\widetilde{\mathcal{G}}^{(t)}$, we can denote the difference as an absent edge $(u,v)$, e.g., $\mathcal{G}^{(t)}$ does not have it but $\widetilde{\mathcal{G}}^{(t)}$ does. Then, according to Eq. 25, the ratio $\Gamma$ of the marginal probability of outputting the same output is in the equation below, where the first term represents that $\mathbb{M}_+^{(t)}$ successfully flips $A(u,v)^{(t)}$ and $\mathbb{M}_-^{(t)}$ can do nothing on the flipped $A(u,v)^{(t)}$ based on our design.

$$\Gamma = \underbrace{\frac{Pr[A(u,v)^{(t)} = 0 \to 1]}{Pr[\widetilde{A}(u,v)^{(t)} = 1 \to 1]}}_{\mathbb{M}_+^{(t)}} \underbrace{\frac{Pr[A(u,v)^{(t)} = 1 \to 1]}{Pr[\widetilde{A}(u,v)^{(t)} = 1 \to 1]}}_{\mathbb{M}_-^{(t)}} + \underbrace{\frac{Pr[A(u,v)^{(t)} = 0 \to 0]}{Pr[\widetilde{A}(u,v)^{(t)} = 1 \to 1]}}_{\mathbb{M}_+^{(t)}} \underbrace{\frac{Pr[A(u,v)^{(t)} = 0 \to 0]}{Pr[\widetilde{A}(u,v)^{(t)} = 1 \to 0]}}_{\mathbb{M}_-^{(t)}}$$

$$= \frac{1/e^{\varepsilon_+^{(t)}}}{1} \times \frac{1}{1 - 1/e^{\varepsilon_-^{(t)}}} + \frac{1 - 1/e^{\varepsilon_+^{(t)}}}{1} \frac{1}{1/e^{\varepsilon_-^{(t)}}} = \frac{e^{\varepsilon_-^{(t)}}}{e^{\varepsilon_+^{(t)}}} \times \frac{1}{e^{\varepsilon_-^{(t)}} - 1} + e^{\varepsilon_-^{(t)}} - \frac{e^{\varepsilon_-^{(t)}}}{e^{\varepsilon_+^{(t)}}} < \frac{e^{\varepsilon_-^{(t)}}}{e^{\varepsilon_+^{(t)}}} \times \frac{1}{e^{\varepsilon_-^{(t)}} - 1} + e^{\varepsilon_-^{(t)}} < \frac{1}{e^{\varepsilon_+^{(t)}}} + e^{\varepsilon_-^{(t)}} \tag{26}$$

Then, we can derive Theorem 4.1 based on Eq. 26. Moreover, we have analyzed the differential privacy scenario that the later mechanism can overwrite (e.g., $\mathbb{M}_-^{(t)}$ flips the edge that $\mathbb{M}_+^{(t)}$ has just flipped) in the next section, and the bound is looser than in Eq. 26 (i.e., larger than $\frac{1}{e^{\varepsilon_+^{(t)}}} + e^{\varepsilon_-^{(t)}}$), which suggests that forbidding the overwrite is a safer way to protect privacy. $\square$

## A.2 ADDITIONAL PROOF FOR THEOREM 4.1

Here, we extend to analyze the scenario when the later perturbation (e.g., $\mathbb{M}_-^{(t)}$) can overwrite the former perturbation (e.g., $\mathbb{M}_+^{(t)}$) and derive the corresponding differential privacy bound.

Similarly, we start from Eq. 25

$$\frac{Pr[\mathbb{M}^{(t)}(\boldsymbol{A}^{(t)}) = \boldsymbol{S}^{(t)}]}{Pr[\mathbb{M}^{(t)}(\widetilde{\boldsymbol{A}}^{(t)}) = \boldsymbol{S}^{(t)}]} \quad \text{for } i,j \in \{1,2,\ldots,n\}$$

$$= \prod_{v_i,v_j} \frac{Pr[\mathbb{M}_+^{(t)}(A(i,j)) = S(i,j)]}{Pr[\mathbb{M}_+^{(t)}(\widetilde{A}(i,j)) = S(i,j)]} \cdot \prod_{v_i,v_j} \frac{Pr[\mathbb{M}_-^{(t)}(A(i,j)) = S(i,j)]}{Pr[\mathbb{M}_-^{(t)}(\widetilde{A}(i,j)) = S(i,j)]} \tag{27}$$

which is a general formula over all possible edges. To be specific, given two adjacent snapshots $\mathcal{G}^{(t)}$ and $\widetilde{\mathcal{G}}^{(t)}$, we can denote the difference as an absent edge $(u, v)$ without loss of generality, e.g., $\mathcal{G}^{(t)}$ does not have it but $\widetilde{\mathcal{G}}^{(t)}$ does. Then, according to Alg. 1 and Eq. 25, the marginal probability $\Gamma$ of outputting the same output is as follows.

$$
\begin{aligned}
\Gamma &= \underbrace{\frac{Pr[A(u,v)^{(t)} = 0 \to 1]}{Pr[\widetilde{A}(u,v)^{(t)} = 1 \to 1]}}_{\mathbb{M}_+^{(t)}} \underbrace{\frac{Pr[A(u,v)^{(t)} = 1 \to 1]}{Pr[\widetilde{A}(u,v)^{(t)} = 1 \to 1]}}_{\mathbb{M}_-^{(t)}} + \\
&\quad \underbrace{\frac{Pr[A(u,v)^{(t)} = 0 \to 1]}{Pr[\widetilde{A}(u,v)^{(t)} = 1 \to 1]}}_{\mathbb{M}_+^{(t)}} \underbrace{\frac{Pr[A(u,v)^{(t)} = 1 \to 0]}{Pr[\widetilde{A}(u,v)^{(t)} = 1 \to 0]}}_{\mathbb{M}_-^{(t)}} + \\
&\quad \underbrace{\frac{Pr[A(u,v)^{(t)} = 0 \to 0]}{Pr[\widetilde{A}(u,v)^{(t)} = 1 \to 1]}}_{\mathbb{M}_+^{(t)}} \underbrace{\frac{Pr[A(u,v)^{(t)} = 0 \to 0]}{Pr[\widetilde{A}(u,v)^{(t)} = 1 \to 0]}}_{\mathbb{M}_-^{(t)}} \\
&= \frac{1/e^{\varepsilon_+^{(t)}}}{1} \times \frac{1 - 1/e^{\varepsilon_-^{(t)}}}{1 - 1/e^{\varepsilon_-^{(t)}}} + \frac{1/e^{\varepsilon_+^{(t)}}}{1} \times \frac{1/e^{\varepsilon_-^{(t)}}}{1/e^{\varepsilon_-^{(t)}}} + \frac{1 - 1/e^{\varepsilon_+^{(t)}}}{1} \frac{1}{1/e^{\varepsilon_-^{(t)}}} \\
&= \frac{2}{e^{\varepsilon_+^{(t)}}} + e^{\varepsilon_-^{(t)}} + e^{\varepsilon_-^{(t)} - \varepsilon_+^{(t)}} < \frac{2}{e^{\varepsilon_+^{(t)}}} + e^{\varepsilon_-^{(t)}}
\end{aligned}
$$

where the first term represents that $\mathbb{M}_+^{(t)}$ successfully flips and $\mathbb{M}_-^{(t)}$ ignores, the second term means that $\mathbb{M}_+^{(t)}$ successfully flips and $\mathbb{M}_-^{(t)}$ also flips and overwrites, and the third terms means that $\mathbb{M}_+^{(t)}$ ignores and $\mathbb{M}_-^{(t)}$ flips.

### A.3 PSEUDO CODES

---

**Algorithm 1** Graph Perturbation based on Differential Privacy

---

**Input:**
> graph snapshot $\mathcal{G}^{(t)}$, privacy budgets $e^{\varepsilon_+^{(t)}}$ and $e^{\varepsilon_-^{(t)}}$

**Output:**
> graph snapshot $\hat{\mathcal{G}}^{(t)}$

1: **for** a certain edge type **do**
> /* Apply $\mathbb{M}_+^{(t)}$ */
2:    **for** edge in non-existing_set **do**
3:       Randomly generate a probability $\alpha$
4:       **if** $\alpha \leq 1/e^{\varepsilon_+^{(t)}}$ **then**
5:          Add edge to set append_set
6:       **end if**
7:    **end for**
> /* Apply $\mathbb{M}_-^{(t)}$ */
8:    **for** edge in existing_set **do**
9:       Randomly generate a probability $\alpha$
10:      **if** $\alpha \leq 1/e^{\varepsilon_-^{(t)}}$ **then**
11:         Add edge to set del_set
12:      **end if**
13:    **end for**
14: **end for**
15: residual_set = set_diff(existing_set, del_set)
16: $\hat{\mathcal{G}}^{(t)}$ = set_union(residual_set, append_set)

---

---

**Algorithm 2** Privacy-Utility Adversarial Training

---

**Input:**
    sampled walks $\{\mathcal{W}\}$ from $\mathcal{G}$, sampled walks $\{\hat{\mathcal{W}}\}$ from $\hat{\mathcal{G}}$, randomly sampled walks $\{\hat{\mathcal{W}}'\}$,
**Output:**
    synthetic walks generator $\mathbb{G}$
  1: **while** until not converge **do**
  2:     Initialize $\mathbb{D}_{util}$, $\mathbb{D}_{prv}$, $\mathbb{G}$
  3:     **for** batches $\mathcal{B}$ in $\{\mathcal{W}\}$, $\mathcal{B}'$ in $\{\mathcal{W}'\}$ **do**
  4:         Calculate $\mathcal{L}_{\mathbb{D}_{util}}(\mathcal{B}, \mathcal{B}')$ in Eq. 10
  5:         Update $\mathbb{D}_{util}$
  6:     **end for**
  7: **end while**
  8: **while** until not converge **do**
  9:     **for** batches $\hat{\mathcal{B}}$ in $\{\hat{\mathcal{W}}\}$ **do**
10:         Initialize noise $z = \{z_i \sim \mathcal{N}(0, 1)\}$
11:         $\mathcal{B}^* = \mathbb{G}(z)$
12:         Calculate $\mathcal{L}_{\mathbb{D}_{prv}}(\hat{\mathcal{B}}, \mathcal{B}^*)$ in Eq. 12
13:         Update $\mathbb{D}_{prv}$
14:         Calculate $\mathcal{L}_{\mathbb{G}}(\mathcal{B}^*)$ in Eq. (12)
15:         Update $\mathbb{G}$
16:     **end for**
17: **end while**

---

## A.4   Theoretical and Empirical Complexity Analysis

**Theoretical Time Complexity Analysis**   For the overall training process, the time complexity of THEPUFF mostly depends on the number of walks sampled (which typically can be approximately $O(E)$ or $O(k|V|)$, $k$ is a constant indicating the number of walks sampled starting from one node, depending on graph sparsity). For each training iteration, the time complexity for forward will be mainly composed of three parts: LSTM with $l$ (i.e., length of walks, typically less than 5) steps, and two tri-level attention networks with time complexity $O(l^2(2d + d_o))$, where $d$ is the dimension of node embedding and $d_o$ is the length of one-hot embedding of the node type. Similarly, the time complexity for graph generation mostly depends on the number of walks to be generated.

Compared with diffusion-based generation methods, our training and generation process both follow a node-by-node sampling procedure, whereas reverse process in current diffusion graph generation models follow an edge-by-edge sampling procedure. Therefore, our work has a complexity linear to $|V|$ while diffusion graph generation models have complexities quadratic to $|V|$ (Table. 1). Although our probability prediction module (tri-level attention layer) might have a higher complexity compared to diffusion models with simpler architecture (e.g., MPNN), if we consider that hyperparameters $(l, d, d_o)$ are much smaller than $|V|$, (empirically, they actually do), our method is still more efficient compared with diffusion-based methods, especially when $|V|$ reaches thousands to even millions.

**Empirical Running Time Comparison**   To give a better illustration from the practical viewpoint, the overall training process of ML-100K given sampled walks (sampling 5 walks of length 5 starting from each node) and initialized nodes' temporal embeddings takes around 350 seconds; the generation process by generating 300,000 walks (which should be much larger than the number of walks needed) takes around 250 seconds. Also note that ML-100k is a cleaned dense dataset, whereas large graphs (which are often sparser) mostly do not require sampling as many as 5 walks from each node.

More importantly, we report the total running time of all baselines generating the temporal heterogeneous graphs for ML-100k dataset in Table 2 below, given diffusion methods are prone to OOM on larger datasets.

Table 5: Running Time Comparison on ML-100k Dataset.

| Methods | Running Time |
|---|---|
| HGEN | ∼ 1120s |
| TagGen | ∼ 2730s |
| TG-GAN | ∼ 4731s |
| EDGE | ∼ 7hrs |
| DISCO | ∼ 3500s |
| GraphMaker | ∼ 4hrs |
| THEPUFF (ours) | ∼ 350s |

**Empirical GPU Storage Comparison** Here, we also report the GPU analysis for all baselines on the ML-100k dataset. For DISCO baseline, its actual GPU usage can be 54GB, verified on a Tesla A100 80 GB machine.

Table 6: GPU Consumption on ML-100k Dataset.

| Methods | GPU Usage |
|---|---|
| HGEN | ∼ 610MB |
| TagGen | ∼ 23GB |
| TG-GAN | ∼ 940MB |
| EDGE | ∼ 12GB |
| DISCO | > 32GB |
| GraphMaker | ∼ 1100MB |
| THEPUFF (ours) | ∼ 712MB |

## A.5 DATASETS AND IMPLEMENTATION DETAILS

Table 7: Temporal Heterogeneous Graph Datasets

| | Type | #Nodes | #Node Types | #Edges | #Edge Types | Time |
|---|---|---|---|---|---|---|
| MovieLens 100k(Harper & Konstan, 2015) | Rating | 2,644 | 4 | 102,625 | 3 | 7 months |
| DBLP(Tang et al., 2008) | Citation | 17,876 | 4 | 51,137 | 4 | 10 years |
| MovieLens 20M(Harper & Konstan, 2015) | Rating | 165,790 | 3 | 20,027,541 | 2 | 21 years |
| Taobao(Zhu et al., 2018) | E-Commerce | 1,009,827 | 3 | 2,932,288 | 2 | 10 hours |

**Datasets**. Detailed dataset statistics are summarized in Table 7. For the DBLP dataset, we retrieve the subgraph by constraining the paper venue as KDD. MovieLens-100k[2], DBLP[3], MovieLens-20M[4], and Taobao[5] are publicly available.

- In MovieLens-100k, node types are user, movie, genre, and occupation; edge types are *user–movie*, *movie–genre*, and *user–occupation*.
- In DBLP, node types are paper, author, field of study, and affiliation; edge types are *author–paper*, *paper–paper*, *author–affiliation*, and *paper–field of study*.
- In MovieLens-20M, node types are user, movie, genre; edge types are *user–movie* and *movie–genre*.
- In Taobao, node types are user, item, and category; edge types are *user–item* and *item–category*.

**Baselines**. Non-temporal baselines (i.e., HGEN and EDGE) are run in a snapshot-to-snapshot way. E.g., for dataset $\mathcal{G} = \{\mathcal{G}^{(1)}, ..., \mathcal{G}^{(T)}\}$ and non-temporal baseline $\mathbb{F}(\cdot)$, $\mathcal{G}^* = \{\mathbb{F}(\mathcal{G}^{(1)}), ..., \mathbb{F}(\mathcal{G}^{(T)})\}$ is considered as the generated temporal heterogeneous graph.

For non-heterogeneous baselines (i.e., TagGen, TG-GAN, and EDGE), node type information is ignored when running baselines. When evaluating performances, node types are considered the same as in the original graph.

---

[2]https://www.kaggle.com/datasets/prajitdatta/movielens-100k-dataset
[3]https://www.aminer.org/citation
[4]https://www.kaggle.com/datasets/grouplens/movielens-20m-dataset
[5]https://tianchi.aliyun.com/dataset/649

**Hyperparameters**. Table 2 is implemented with the following hyperparameters:

- $\epsilon_- = 8$ for all datasets, $\epsilon_+$ is decided by Eq. 4.
- batch size = 32 for MovieLens 100K dataset and DBLP dataset, 64 for other datasets;
- node embedding dimension = 128;
- hidden dimensions are all set to 128;
- dropout rate = 0.2 in the attention layer;
- learning rate = $1e-4$ for the generator and $1e-3$ for discriminators;
- SGD optimizer is used for discriminators, while RMSprop optimizer is used for the generator;
- JUST (Hussein et al., 2018) is applied to initialize node embeddings. In the running of JUST, we have the maximum walk length as 100; sample maximum of 10 walks starting from each node.

Hyperparamters for HGEN, TagGen and TG-GAN are set as the default or recommended value in their public code repositories. Hyperparameters for EDGE are set to recommended values except that batch size is set to 1 due to memory limitation.

**Machine Configuration**. All experiments are performed on a Linux platform with Intel(R) Xeon(R) Gold 6240R CPU and Tesla V100 SXM2 32GB GPU.

**Reproducibility**. The code will be published on the authors' websites upon the paper's publication.

### A.6  CONVERGENCE OF THEPUFF TRAINING

In this section, we discuss the convergence of the adversarial training part of THEPUFF (lines $8-17$ in Alg. 4). In practice, our generator $\mathbb{G}$ is updated every 2-5 iterations while privacy discriminator $\mathbb{D}_{prv}$ is updated every iteration. An illustration of the training process on the largest dataset (Taobao) is attached in Figure 3. In the training process, we first see a decrease of $\mathcal{L}_{\mathbb{D}_{prv}}$ to the lowest point, and then an increase of $\mathcal{L}_{\mathbb{D}_{prv}}$ as $\mathbb{G}$ gradually starting to outperform $\mathbb{D}_{prv}$.

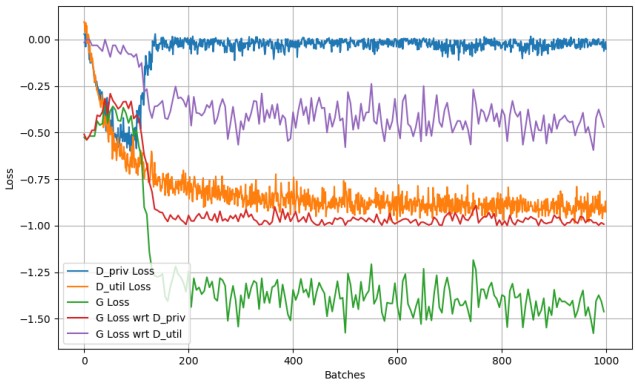

Figure 3: Training curve of generator/discriminators on Taobao

### A.7  ABLATION STUDY

Here, we also conduct an ablation study on the dataset MovieLens 100k and MovieLens 20M. The results are shown in Table 8, where we trained our framework without either the utility-preserving discriminator ($\mathbb{D}_{util}$) or the privacy-guaranteed discriminator ($\mathbb{D}_{prv}$). It can be observed that with only the generator and utility-preserving discriminator (i.e., w/o $\mathbb{D}_{prv}$), our THEPUFF can also generate the comparable privacy-guaranteed synthetic graph, which proves our design in Eq. 13 that the nodes are sampled from the perturbed graph $\hat{\mathcal{G}}$, thereby privacy-preserved information obtained from differential privacy perturbation $\mathbb{M}$ is transferred to the generator $\mathbb{G}$.

Table 8: Ablation Study Experiment Results.

| Datasets | Methods | Utility | | | | | | Privacy |
|----------|---------|---------|---|---|---|---|---|---------|
| | | Cluster ($\downarrow$) | TC ($\downarrow$) | LCC ($\downarrow$) | Degree ($\downarrow$) | Meta-2 ($\downarrow$) | Meta-3 ($\downarrow$) | EO-Rate ($\downarrow$) |
| ML-100k | w/o $\mathbb{D}_{util}$ | $0.468_{(\pm 0.018)}$ | $1.000_{(\pm 4.361e^{-4})}$ | $0.240_{(\pm 4.361e^{-4})}$ | $0.079_{(\pm 3.566e^{-3})}$ | $1.660_{(\pm 3.928e^{-4})}$ | $1.313_{(\pm 8.650e^{-6})}$ | $2.319e^{-2}_{(\pm 2.302e^{-4})}$ |
| | w/o $\mathbb{D}_{prv}$ | $0.496_{(\pm 0.008)}$ | $1.050_{(\pm 1.308e^{-3})}$ | $0.216_{(\pm 2.294e^{-2})}$ | $0.073_{(\pm 4.564e^{-2})}$ | $1.660_{(\pm 6.067e^{-4})}$ | $1.313_{(\pm 1.500e^{-6})}$ | $2.068e^{-2}_{(\pm 2.059e^{-3})}$ |
| ML-20M | w/o $\mathbb{D}_{util}$ | $0.0_{(\pm 0.0)}$ | $0.0_{(\pm 0.0)}$ | $0.874_{(\pm 1.262e^{-1})}$ | $0.560_{(\pm 1.589e^{-2})}$ | $1.461_{(\pm 6.321e^{-2})}$ | $1.266_{(\pm 0.0)}$ | $5.501e^{-4}_{(\pm 5.501e^{-4})}$ |
| | w/o $\mathbb{D}_{prv}$ | $0.0_{(\pm 0.0)}$ | $0.0_{(\pm 0.0)}$ | $0.512_{(\pm 4.879e^{-1})}$ | $0.498_{(\pm 7.542e^{-2})}$ | $1.551_{(\pm 3.910e^{-5})}$ | $1.266_{(\pm 2.000e^{-7})}$ | $0.0_{(\pm 0.0)}$ |

## A.8 ATTACK EXPERIMENT ON ALL DATASETS

Here, we follow the attacker model as expressed in Eq.23 and test the privacy protection effectiveness of baseline methods across all datasets. The results are shown below, where we can see the proposed THEPUFF method can reduce the successful attack probability to the largest extent.

Table 9: Attack Experiment on All Datasets.

| | Successful Attack Probability (%) ($\downarrow$) | | | |
|-------------------|---------|-------|---------|--------|
| Baseline / Dataset | ML-100k | DBLP | ML-20M | Taobao |
| Original | 4.192 | 0.093 | 0.352 | 0.006 |
| HGEN | $0.037_{(0.001)}$ | $0.009_{(1.3e^{-4})}$ | $0.007_{(0.005)}$ | $1.3e^{-4}_{(2.0e^{-7})}$ |
| DPGGAN | $0.037_{(3.6e^{-4})}$ | $0.008_{(1.3e^{-4})}$ | OOM | OOM |
| THEPUFF(Ours) | $\mathbf{0.021_{(0.007)}}$ | $\mathbf{0.006_{(9e^{-5})}}$ | $\mathbf{7.5e^{-4}_{(6.0e^{-5})}}$ | $\mathbf{7.9e^{-5}_{(6.2e^{-6})}}$ |

## A.9 ATTACKING USERS/AUTHORS ON ALL DATASETS

Here, we design a more complicated attacker model by extending the attacker in Eq.23 as follows. For a specific node type $o$ of attacker's interest,

$$p_i^{(t)} = \begin{cases} 0 & \text{, otherwise} \\ \frac{1}{\sum_{v \in \mathcal{G}^*} \mathbb{1}[d_{gen}^{(t)}[v]==d_{gen}^{(t)}[i] \text{ and } \phi_{gen}^{(t)}[v]==o]} & \text{, } d_{gen}^{(t)}[i] = d_{ori}^{(t)}[i] \text{ and } \phi_{gen}^{(t)}[i] = o \end{cases} \quad (28)$$

where the added $\phi_{gen}^{(t)}[i]$ means the node type of node $i$ at time $t$ on the generated graph.

Therefore, we can now test how the baselines protect the privacy of certain types of entities, e.g., internet users in ML-100K dataset, ML-20M dataset, Taobao datasets, and authors in DBLP datasets.

The results are shown in the table below, where our proposed THEPUFF also achieves the best performance across all datasets.

Table 10: Attacking Users/Authors on All Datasets.

| | Successful Attack Probability (%) ($\downarrow$) | | | |
|-------------------|---------|-------|---------|--------|
| Baseline / Dataset | ML-100k | DBLP | ML-20M | Taobao |
| Original | 11.73 | 0.113 | 0.427 | 0.021 |
| HGEN | $0.108_{(0.008)}$ | $0.013_{(5e^{-6})}$ | $9.3e^{-4}_{(7.2e^{-6})}$ | $5.5e^{-4}_{(1.4e^{-5})}$ |
| DPGGAN | $0.095_{(0.004)}$ | $0.024_{(1.5e^{-5})}$ | OOM | OOM |
| THEPUFF(Ours) | $\mathbf{0.095_{(0.002)}}$ | $\mathbf{0.007_{(8.2e^{-4})}}$ | $\mathbf{6.9e^{-4}_{(8.6e^{-6})}}$ | $\mathbf{3.6e^{-4}_{(2.4e^{-5})}}$ |

## A.10 PSEUDO-CODE OF $\mathbb{D}_{util}$ AND $\mathbb{D}_{prv}$

Here we show the details of calculating $\mathbb{D}_{util}$ and $\mathbb{D}_{prv}$ using pseudo-code, i.e., how $\mathbb{D}_{util}$ calculate the probability of input walk being sampled from real graph, and how $\mathbb{D}_{prv}$ calculate the probability of input walk being sampled from perturbed graph.

As $\mathbb{D}_{util}$ and $\mathbb{D}_{prv}$ share the exact same architecture, we only show the pseudo-code of $\mathbb{D}_{util}$, pseudo-code for $\mathbb{D}_{prv}$ is exactly a copy of $\mathbb{D}_{util}$.

In detail, for a sampled walk, we first fetch the node embeddings of the nodes in the walk (line 1). Then, the node embeddings and the sampled walk are fed into the tri-level attention layer (line 2 – 3), followed by a fully connected layer (line 4 – 5) where probability of the sampled walk being sampled from original graph (or perturbed graph for $\mathbb{D}\_prv$) is the output.

---

**Algorithm 3** Pseudo-code of $\mathbb{D}_{util}()$

---

**Input:**

Sampled walk $\mathcal{W}_i = \{\{v_1^{(t)}, v_2^{(t)}, ..., v_l^{(t)}\}, \{o_1, o_2, ..., o_l\}\}$; node embedding vectors $\{h_i^{(t)}\}$; tri-level attention layer as described in Eq. 6 – 9; a fully connected layer with weight matrix $\mathbf{W}_O \in \mathcal{R}^{d \times 2}$ and biases $\mathbf{b} \in \mathcal{R}^2$.

**Output:**

Probability that $\mathcal{W}_i$ is sampled from the real graph.

1: Fetch embeddings $\mathbf{H} \in \mathcal{R}^{l \times d}$, where i-th row $h_{v_i}^{(t)}$ is the embedding of $v_i^{(t)}$
2: Compute tri-level attention scores $\mathbf{A}^{time}, \mathbf{A}^{node}, \mathbf{A}^{type}$ through Eq. 6 – 8
3: Calculate output of attention layer $\mathbf{Z}$ in Eq. 9
4: Calculate 2-dimensional output of fully connected layer $[p_0, p_1]^T = \mathbf{Z}\mathbf{W}_O + \mathbf{b}$
5: Output probability $\frac{e^{p_0}}{e^{p_0} + e^{p_1}}$

---

### A.11 PSEUDO-CODE OF ASSEMBLER

To better explain how assembler handle the temporal information, pseudo-code for assembling temporal heterogeneous graph from scoring matrix and meta-path frequencies is shown below. Note that time superscripts $^{(t)}$ of nodes are emitted in Eq. 16 and Eq. 17 for brevity.

In detail, after we sample a sufficient amount of temporal heterogeneous random walk from generator, we count the frequencies of edges and meta-paths to get scoring matrix $\mathbf{S}$ and meta-path frequencies $p_O$. With these two as inputs, assembler iteratively assembles the generated temporal heterogeneous graph as follows. In each iteration, we first sample a starting node based on node degrees in the scoring matrix (line 2). Then, we sample a meta-path that starts with the starting node's node type based on meta-path frequencies (line 3). Finally, we iteratively sample the rest of the nodes in the sampled meta-path one by one and add the sampled edges to the assembled graph (line 4 – 8): the subsequent node is sampled based on next node type and scoring matrix (line 5), and the edge between the sampled subsequent node and current node is added to the assembled graph (line 6 – 7). Finally, after we get desired amount of edges, the sampling iteration (line 1 – 9) stops and output the final assembled temporal heterogeneous graph.

---

**Algorithm 4** Pseudo-code of Assembler

---

**Input:**

Scoring matrix $\mathbf{S}$; meta-path frequencies $p_O$.

**Output:**

Assembled temporal heterogeneous graph $\mathcal{G} = \{\mathcal{G}^{(1)}, \mathcal{G}^{(2)}, \dots, \mathcal{G}^{(T)}\}$.

1: **while** not enough edges **do**
2:    Sample a starting node $v_1^{(t_1)}$ based on probability in Eq. 16.
3:    Sample a meta-path $\mathbf{O} = (o_1, ...o_l) \sim p_O$
4:    **for** i in $1, 2, ..., l - 1$ **do**
5:       Sample $v_{i+1}^{(t_{i+1})}$ based on probability in Eq. 17
6:       Add edge $(v_i^{(t_i)}, v_{i+1}^{(t_i)})$ to $\mathcal{G}^{(t_i)}$
7:       Add edge $(v_i^{(t_{i+1})}, v_{i+1}^{(t_{i+1})})$ to $\mathcal{G}^{(t_{i+1})}$
8:    **end for**
9: **end while**

---

