# OpenReview forum: "Temporal Heterogeneous Graph Generation with Privacy, Utility, and Efficiency"
_ICLR.cc/2025/Conference — ICLR 2025 Spotlight_

### Official Review · Reviewer_ynGq · 2024-10-28

**Soundness:** 3
**Presentation:** 3
**Contribution:** 2
**Rating:** 6
**Confidence:** 3

**Summary:**

This paper introduces an innovative model, THEPUFF, for temporal heterogeneous graph generation that balances privacy protection and data utility. By leveraging adversarial learning and differential privacy perturbation, THEPUFF effectively safeguards privacy while preserving usability, with comprehensive experiments demonstrating its robustness and utility. Suggestions for improvement include clarifying methodological details, adding comparative experiments with existing methods, incorporating additional attack tests, and providing efficiency analysis. Overall, THEPUFF shows strong practical potential and innovation, and further optimization could make significant contributions to privacy protection in relational data.

**Strengths:**

1. The use of adversarial learning and differential privacy perturbation is well-justified, and the paper provides six metrics to comprehensively evaluate the model’s effectiveness in privacy and utility preservation.
2. The model is designed for applications that require scalable, privacy-sensitive graph data generation, with potential uses in recommendation systems, question answering, and beyond.
3. Extensive experiments, including attack simulations, ablation studies, and parameter analyses, validate the robustness and practicality of the model across diverse scenarios.

**Weaknesses:**

1. The introduction references social networks and citation networks as examples, but does not provide specific data or case studies to illustrate the severity of the issues and their potential impacts, making the discussion appear abstract.
2. While THEPUFF claims efficiency, the paper lacks explicit efficiency metrics or runtime comparisons, which would substantiate this claim and assess scalability
3. The model’s privacy resilience could be more comprehensively assessed by testing against a broader range of privacy attacks, such as structural or attribute inference attacks.

**Questions:**

See the weaknesses above

---

> ### Author Response · Authors · 2024-11-25
> **Reply from Authors (Part I)**
>
> We are quite grateful for your appreciation of our paper’s motivation, potential, and comprehensive experiments.
>
> Also, thanks very much for your questions and suggestions. Addressing them improves the paper’s quality to the next level. Each detailed action and answer are presented in the below Q&A format.
>
> > **W1**. The introduction references social networks and citation networks as examples, but does not provide specific data or case studies to illustrate the severity of the issues and their potential impacts, making the discussion appear abstract.
>
> In the introduction, the toy example aims to be an illustration to demonstrate the existence of the privacy issue.
>
> In the main paper (i.e., Section 5.3), we presented the formal definition of an attacker in its mathematical form who knows the temporal structural background knowledge and uses it to identify victims.
> - According to Table 3 in Section 5.3, we can see, in the original graph, if no protection mechanism is added, the attacker has the successful attack probability to be 0.00093. Adding our method, the same attacker only has the probability of 0.00006, which probability is decreased by 93.15% confidence and it is the largest decrease compared to other baselines.
>
> To better address the your concern, during this rebuttal,
>
> - **First**, we have extended our attack experiments in Section 5.3 to all four datasets, and the results are placed in Appendix A.8 with Table 7. In this table, we can see that our method still achieves the best performance across all datasets than all baselines. For your information, we put the table below. More detailed experimental description and analysis can be found in Appendix A.8, and the new paper is uploaded with new contents in the yellow color background.
>
> Table. Attack Experiment  on All Datasets. Successful Attack Probability (\%) ($\downarrow$)
> | Baseline / Dataset     | ML-100k             | DBLP               | ML-20M               | Taobao               |
> |--------------|---------------------|--------------------|----------------------|----------------------|
> | Original | 4.192 | 0.093 | 0.352 | 0.006               |
> | HGEN     | $0.037_{(0.001)}$ | $0.009_{(1.3e^{-4})}$ | $0.007_{(0.005)}$ | $1.3e^{-4}_{(2.0e^{-7})}$    |
> | DPGGAN   |  $0.037_{(3.6e^{-4})}$ | $0.008_{(1.3e^{-4})}$ | OOM | OOM                |
> | THePUff (Ours)     |  $\mathbf{0.021_{(0.007)}}$ | $\mathbf{0.006_{(9e^{-5})}}$ | $\mathbf{7.5e^{-4}_{(6.0e^{-5})}}$ | $\mathbf{7.9e^{-5}_{(6.2e^{-6})}}$  |
>
> The above table with detailed description and discussion is added to Appendix A.8 within the yellow background.
>
> - **Second**, we designed a new powerful attacker, which is more powerful in the temporal heterogeneous setting and can narrow down the candidate scope by distinguishing the node type. Its mathematical modeling is expressed in Eq. 28.
>   - Different from the previous attacker in Eq. 23, when the node type is the same and the structural role is also the same at a certain timestamp, then the victim-candidate relationship can be established.
>   -  For example, it means that, in the citation network, an author who wrote 6 papers and a paper which is cited 6 times are not regarded as the same candidate by the attacker, only two authors who wrote the same paper at a certain time period will be regarded as the target.
>
> The new experimental results across all datasets are shown below, where we can see our proposed method still performs the best in terms of privacy protection.
>
> Table. Attack Experiment of Target Types on All Datasets. Successful Attack Probability (\%) ($\downarrow$)
> | Baseline / Dataset      | ML-100k             | DBLP               | ML-20M               | Taobao               |
> |--------------|---------------------|--------------------|----------------------|----------------------|
> | Original | 11.73              | 0.113              | 0.427                | 0.021               |
> | HGEN     | 0.108 (0.008)      | 0.013 (5e⁻⁶)       | 9.3e⁻⁴ (7.2e⁻⁶)     | 5.5e⁻⁴ (1.4e⁻⁵)    |
> | DPGGAN   | 0.095 (0.004)      | 0.024 (1.5e⁻⁵)     | OOM                  | OOM                 |
> | THePUff (Ours)     | **0.095 (0.002)**  | **0.007 (8.2e⁻⁴)** | **6.9e⁻⁴ (8.6e⁻⁶)** | **3.6e⁻⁴ (2.4e⁻⁵)** |
>
> The above table with detailed description and discussion is added to Appendix A.9 within the yellow background.

---

> > ### Author Response · Authors · 2024-11-25
> > **Reply from Authors (Part II)**
> >
> > > **W2**. While THEPUFF claims efficiency, the paper lacks explicit efficiency metrics or runtime comparisons, which would substantiate this claim and assess scalability
> >
> > **From the theoretical perspective**,
> >
> > - We claim that Transformer-based Generative Adversarial Networks like ThePuff are more efficient compared to diffusion-based generation methods, because complexity of diffusion-based methods is generally quadratic of number of nodes while complexity of GAN can be linear of number of nodes, with the theoretical complexity of different diffusion methods listed in Table 1.
> >
> > - During this rebuttal, we also added the detailed analysis about why the theoretical time complexity of THePUff is superior than diffusion methods in Appendix A.4, marked within the yellow background.
> >
> > **From the empirical perspective**,
> >
> > - To give a direct comparison between THePUff and the diffusion models and other baselines, we record their running time and GPU memory cost on the ML-100k dataset, given diffusion models costs OOM on middle and large datasets.
> >
> > - The comprehensive comparison is shown below. Also, compared with other non-diffusion generation baselines, TagGen, TG-GAN and HGEN, no matter in running or GPU consumption, our proposed THePUff achieves the leading position.
> >
> > Table. Running Time Comparison on ML-100k Dataset
> > | Methods      | Running Time |
> > |--------------|--------------|
> > | HGEN         | ~1120s       |
> > | TagGen       | ~2730s       |
> > | TG-GAN       | ~4731s       |
> > | EDGE         | ~7hrs        |
> > | DISCO        | ~3500s       |
> > | GraphMaker   | ~4hrs        |
> > | THePUff (ours)  | ~350s        |
> >
> > Table. GPU Consumption on ML-100k Dataset
> > | Methods      | GPU Usage  |
> > |--------------|------------|
> > | HGEN         | ~610MB     |
> > | TagGen       | ~23GB      |
> > | TG-GAN       | ~940MB     |
> > | EDGE         | ~12GB      |
> > | DISCO        | >32GB      |
> > | GraphMaker   | ~1100MB    |
> > | THePUff (ours)  | ~712MB     |
> >
> > The above two tables along with detailed discussion and analysis are added in Appendix A.4 within the yellow background.

---

> > > ### Author Response · Authors · 2024-11-25
> > > **Reply from Authors (Part III)**
> > >
> > > > **W3**. The model’s privacy resilience could be more comprehensively assessed by testing against a broader range of privacy attacks, such as structural or attribute inference attacks.
> > >
> > > To further address your concern, we made two kinds of efforts during this rebuttal.
> > > **First**, we have extended the attack experiments on all datasets, and the observation is consistent and robust with the previous results, i.e., our proposed method can reduce the attacker’s successful probability to the largest extent, compared to SOTA, as shown below.
> > >
> > > Table. Attack Experiment on All Datasets. Successful Attack Probability (\%) ($\downarrow$)
> > > | Dataset      | ML-100k               | DBLP                  | ML-20M                  | Taobao                  |
> > > |--------------|-----------------------|-----------------------|-------------------------|-------------------------|
> > > | Original | 4.192                | 0.093                | 0.352                   | 0.006                  |
> > > | HGEN     | 0.037 (0.001)        | 0.009 (1.3e⁻⁴)       | 0.007 (0.005)           | 1.3e⁻⁴ (2.0e⁻⁷)       |
> > > | DPGGAN   | 0.037 (3.6e⁻⁴)       | 0.008 (1.3e⁻⁴)       | OOM                     | OOM                    |
> > > | THePUff (Ours)    | **0.021 (0.007)**    | **0.006 (9e⁻⁵)**     | **7.5e⁻⁴ (6.0e⁻⁵)**    | **7.9e⁻⁵ (6.2e⁻⁶)**   |
> > >
> > > The above table with detailed description and analysis has been added to the paper in Appendix A.8 as Table 7, marked by the yellow background.
> > >
> > >
> > > **Second**, we have designed a new experiment that is even more challenging and more close to the realistic scenarios.
> > >
> > > In our previous attacker model, as expressed in Eq. 23, the attacker will not distinguish the node types, which means that as long as the structural role is the same, the nodes will be identified as the victim candidate. For example, an author who wrote 6 papers and a paper which is cited 6 times are regarded as the same by the attacker.
> > >
> > > To consider a more complex and strong attacker, we empower the attacker by allowing it to know more background knowledge, which means only when the node type is the same and the structural role is the same, then the victim-candidate relationship can be established.
> > >
> > > - The mathematical details are added as Eq. 28 in the updated paper. For example, it means that, in the citation network, an author who wrote 6 papers and a paper which is cited 6 times are not regarded as the same candidate by the attacker, only two authors who wrote the same paper at a certain time period will be regarded as the target.
> > >
> > > The new experimental results across all datasets are shown below, where we can see our proposed method still performs the best in terms of privacy protection.
> > >
> > > Table. Attack Experiment of Target Types on All Datasets. Successful Attack Probability (\%) ($\downarrow$)
> > > | Baseline / Dataset      | ML-100k             | DBLP               | ML-20M               | Taobao               |
> > > |--------------|---------------------|--------------------|----------------------|----------------------|
> > > | Original | 11.73              | 0.113              | 0.427                | 0.021               |
> > > | HGEN     | 0.108 (0.008)      | 0.013 (5e⁻⁶)       | 9.3e⁻⁴ (7.2e⁻⁶)     | 5.5e⁻⁴ (1.4e⁻⁵)    |
> > > | DPGGAN   | 0.095 (0.004)      | 0.024 (1.5e⁻⁵)     | OOM                  | OOM                 |
> > > | THePUff (Ours)     | **0.095 (0.002)**  | **0.007 (8.2e⁻⁴)** | **6.9e⁻⁴ (8.6e⁻⁶)** | **3.6e⁻⁴ (2.4e⁻⁵)** |
> > >
> > > The above table with detailed description and analysis has been added to the paper in Appendix A.9 as Table 8, marked by the yellow background.

---

> > > > ### Comment · Reviewer_ynGq · 2024-11-26
> > > >
> > > > Thank you for your reply. I would keep my score.

---

### Official Review · Reviewer_jsYy · 2024-11-04

**Soundness:** 3
**Presentation:** 3
**Contribution:** 3
**Rating:** 8
**Confidence:** 4

**Summary:**

The introduces THEPUFF, a framework for generating temporal heterogeneous graphs that balances privacy, utility, and efficiency. THEPUFF employs a differential privacy algorithm to perturb input graphs, protecting privacy, and then uses a generative adversarial setting to learn and produce synthetic graph data that preserves both privacy and utility. The authors propose six new metrics to measure the utility and privacy of the generated graphs in a temporal setting. Extensive experiments on large-scale datasets demonstrate THEPUFF's ability to generate utilizable graphs with protected privacy compared to state-of-the-art baselines. The framework includes a privacy-preserving graph sample for distribution, an adversarial generative model for fitting the distribution, and an assembler for aggregating generated walks into a final graph. THEPUFF's effectiveness is validated through comprehensive evaluations and attacker experiments, showing it generates graphs with lower successful attack probabilities, indicating stronger privacy protection. The paper concludes that THEPUFF is the first work to address temporal heterogeneous graph generation with guaranteed privacy.

**Strengths:**

1. The paper addresses a significant gap in the field by proposing a method for generating temporal heterogeneous graphs with a focus on privacy preservation, which is increasingly important in data-sensitive applications.
2. THEPUFF is designed to balance three key aspects: privacy, utility, and efficiency, which are often competing considerations in data generation tasks. This balance is achieved through a sophisticated approach that includes differential privacy algorithms and generative adversarial networks (GANs).
3. The paper provides a thorough experimental evaluation using multiple real-world datasets, demonstrating THEPUFF's effectiveness and scalability compared to baselines using various metrics.
4. The paper is generally good written and easy to follow.

**Weaknesses:**

1. As far as I am concerned, the key challenge in generating temporal graphs lies in the potential privacy leak in the temporal sense. The paper does not emphasize enough how such a challenge is tackled in the proposed method.
2. The paper carried out comparative experiments against baselines like EDGE (Chen et al. 2023), while excluding other state-of-the-art baselines like DISCO (Xu et al. 2024). Justifications should be provided to the choice of baselines.
3. Some technical details are missing, like the functional form of D_priv, which hampers a smooth understanding of the paper.

**Questions:**

1. How is D_priv calculated in the proposed method? Any parameters involved? How does the parameter setting impact the performance of the proposed method?
2. How does the paper handle the potential privacy leak via temporal relevance in the graph when designing the proposed method? Is this taken care of in the GAN part or the assembly part?
3. Can the paper provide more experimental results against DISCO (Xu et al. 2024), even in small-scale graphs? This could benefit the understanding of the pros and cons of different approaches.

---

> ### Author Response · Authors · 2024-11-25
> **Reply from Authors (Part I)**
>
> We want to express our thanks to your review and your appreciation of our research’s potential, the considerable design of the method, and the thorough experimental evaluation.
>
> Your raised suggestions are very actionable and we took each of them seriously with the answer below.
>
> > **Q1**. How is D_priv calculated in the proposed method? Any parameters involved? How does the parameter setting impact the performance of the proposed method?
>
> To better understand the calculation of $\mathbb{D}\_{prv}$ and $\mathbb{D}\_{util}$, we add the detailed pseudo-code in Appendix A.10.
>
> In Appendix A.10, we show the details of calculating $\mathbb{D}\_{util}$ and $\mathbb{D}\_{prv}$ using pseudo-code, i.e., how $\mathbb{D}\_{util}$ calculate the probability of input walk being sampled from real graph, and how $\mathbb{D}\_{prv}$ calculate the probability of input walk being sampled from perturbed graph.
>
> $\mathbb{D}\_{util}$ and $\mathbb{D}\_{prv}$ share the exact same architecture. For example, in $\mathbb{D}\_{util}$, for a sampled walk, we first fetch the node embeddings of the nodes in the walk. Then, the node embeddings and the sampled walk are fed into the tri-level attention layer, followed by a fully connected layer, where probability of the sampled walk being sampled from original graph (or perturbed graph for $D\_{prv}$) is the output.
>
> **The detailed description is added to the Appendix A.10 within the yellow background, and the new paper is uploaded**.
>
> The hyperparameters and their values involved are also discussed in Appendix A.5, which are common hyperparameters in transformers and linear layers (e.g., batch size, learning rate, embedding size).

---

> ### Author Response · Authors · 2024-11-25
> **Reply from Authors (Part II)**
>
> > **Q2**. How does the paper handle the potential privacy leak via temporal relevance in the graph when designing the proposed method? Is this taken care of in the GAN part or the assembly part?
>
> The privacy leakage can originate from multiple aspects, the temporal dependency is only one of them, and other sources can be structural background knowledge (e.g., who you connects reveal your identity to some extent) and feature background knowledge (e.g., user profile in the social network).
>
> Suppose an attacker attacks the temporal heterogeneous social network to reveal someone’s identity, then our goal is to protect the user’s identity from being revealed at this timestamp.
>
> Therefore, for the protection, our method dives in each timestamp and analyzes the node structural protection and node feature protection.
>
> To be specific, the protection is taken care of in both GAN and assembler parts.
> - GAN part is mainly for the structural protection, because differential-privacy-based permuted temporal heterogeneous graph by Eq. 2 and 3 are served as input for the discriminator, such that the generator can learn how to generated privacy-enhanced graphs without touching any real data.
> - The assembler is mainly responsible for node feature protection, as shown in Algorithm 4 and Eq. 17, the assembler introduces randomness in the sampling process, which follows the meta-path frequency, preserves the locality but randomizes the high-order subgraph-level connection.
>
> During this rebuttal, we also prepared comprehensive attacker experiments to show that the privacy is largely protected by our proposed method.
>
> - **First**, we have extended our attack experiments in Section 5.3 to all four datasets, and the results are placed in Appendix A.8 with Table 7. In this table, we can see that our method still achieves the best performance across all datasets than all baselines. For your information, we put the table below. More detailed experimental description and analysis can be found in Appendix A.8, and the new paper is uploaded with new contents in the yellow color background.
>
> Table. Attack Experiment  on All Datasets. Successful Attack Probability (\%) ($\downarrow$)
> | Baseline / Dataset     | ML-100k             | DBLP               | ML-20M               | Taobao               |
> |--------------|---------------------|--------------------|----------------------|----------------------|
> | Original | 4.192 | 0.093 | 0.352 | 0.006               |
> | HGEN     | $0.037_{(0.001)}$ | $0.009_{(1.3e^{-4})}$ | $0.007_{(0.005)}$ | $1.3e^{-4}_{(2.0e^{-7})}$    |
> | DPGGAN   |  $0.037_{(3.6e^{-4})}$ | $0.008_{(1.3e^{-4})}$ | OOM | OOM                |
> | THePUff (Ours)     |  $\mathbf{0.021_{(0.007)}}$ | $\mathbf{0.006_{(9e^{-5})}}$ | $\mathbf{7.5e^{-4}_{(6.0e^{-5})}}$ | $\mathbf{7.9e^{-5}_{(6.2e^{-6})}}$  |
>
> The above table with detailed description and discussion is added to Appendix A.8 within the yellow background.
>
> - **Second**, we designed a new powerful attacker, which is more powerful in the temporal heterogeneous setting and can narrow down the candidate scope by distinguishing the node type. Its mathematical modeling is expressed in Eq. 28.
>   - Different from the previous attacker in Eq. 23, when the node type is the same and the structural role is also the same at a certain timestamp, then the victim-candidate relationship can be established.
>   -  For example, it means that, in the citation network, an author who wrote 6 papers and a paper which is cited 6 times are not regarded as the same candidate by the attacker, only two authors who wrote the same paper at a certain time period will be regarded as the target.
>
> The new experimental results across all datasets are shown below, where we can see our proposed method still performs the best in terms of privacy protection.
>
> Table. Attack Experiment of Target Types on All Datasets. Successful Attack Probability (\%) ($\downarrow$)
> | Baseline / Dataset      | ML-100k             | DBLP               | ML-20M               | Taobao               |
> |--------------|---------------------|--------------------|----------------------|----------------------|
> | Original | 11.73              | 0.113              | 0.427                | 0.021               |
> | HGEN     | 0.108 (0.008)      | 0.013 (5e⁻⁶)       | 9.3e⁻⁴ (7.2e⁻⁶)     | 5.5e⁻⁴ (1.4e⁻⁵)    |
> | DPGGAN   | 0.095 (0.004)      | 0.024 (1.5e⁻⁵)     | OOM                  | OOM                 |
> | THePUff (Ours)     | **0.095 (0.002)**  | **0.007 (8.2e⁻⁴)** | **6.9e⁻⁴ (8.6e⁻⁶)** | **3.6e⁻⁴ (2.4e⁻⁵)** |
>
> The above table with detailed description and discussion is added to Appendix A.9 within the yellow background.

---

> ### Author Response · Authors · 2024-11-25
> **Reply from Authors (Part III)**
>
> > **Q3**. Can the paper provide more experimental results against DISCO (Xu et al. 2024), even in small-scale graphs? This could benefit the understanding of the pros and cons of different approaches.
>
> **First**, we would like to clarify why we choose EDGE but not DISCO for the submission.
>
> - EDGE is a also graph diffusion generation model which enjoys a lower complexity as it applies node filtering at each diffusion step
> - EDGE is from last year, and so far, it is the one of the state-of-the-art of unsupervised single graph generation models with diffusion, to the best of our knowledge
> - DISCO just published in September this year, and the code is not published yet. Moreover, DISCO is designed for small graphs (e.g., protein or molecular), not quite suitable for the scale-up settings.
>
> **Second**, to better address your concern, during this rebuttal, we contacted the authors of DISCO and finally got the code to execute experiments.
>
> **Third**, we searched the recently published papers and found another graph diffusion work, GraphMaker [1], published in October this year, and included it into our baseline.
>
> With two new baselines, we did the new experiments on all datasets with our previous baselines.
>
> The detailed analysis and comparison are marked by different colors, and they are updated in the paper and within the yellow background. More details can be seen in Table 2 of the paper, **and we make the necessary highlights below**.
>
> - Diffusion models can produce some competitive performance for some metrics, but the balancing of utility, privacy, and efficiency is not as good as the proposed THePUff. For example, as shown in the below table, for the small dataset ML-100k, DISCO can produce similar privacy-protection results with ours but with the sacrifice of utility.  GraphMaker can produce better utilizable generation, but the privacy is not guaranteed.
>
> - Also, both DISCO and GraphMaker caused out-of-memory on middle (DBLP)  and large (ML-20M and Taobao) datasets, which aligns our discussion and analysis in Table 1. For example, DISCO needs 54GB GPU memory for the small ML-100k dataset.
>
> - The running time and GPU consumption of all baselines are also added to Appendix A.4 within the yellow background, where our method outperforms consistently.
>
> ---
>
> | ML-100k       | Cluster                              | TC            | LCC                              | Degree                             | Meta-2                             | Meta-3                              | EO-Rate                              |
> |---------------|---------------------------------------|---------------------|---------------------------------------|---------------------------------------|---------------------------------------|---------------------------------------|---------------------------------------|
> | HGEN          | $5.871e^{-2}_{(\pm6.971e^{-4})}$      | $1.000_{(\pm0.0)}$  | $1.905e^{-1}_{(\pm0.0)}$              | $1.395e^{-1}_{(\pm1.353e^{-3})}$      | $1.469_{(\pm1.850e^{-4})}$            | $1.289_{(\pm4.450e^{-5})}$            | $3.200e^{-2}_{(\pm2.008e^{-4})}$      |
> | TagGen        | $3.021e^{-1}_{(\pm1.561e^{-1})}$      | $1.000_{(\pm0.0)}$  | $6.417e^{-1}_{(\pm0.023)}$            | $1.570e^{-1}_{(\pm5.345e^{-3})}$      | $1.500_{(\pm1.306e^{-1})}$            | $1.310_{(\pm3.281e^{-3})}$            | $2.238e^{-1}_{(\pm1.159e^{-1})}$      |
> | TG-GAN        | $4.987e^{-1}_{(\pm7.305e^{-2})}$      | $1.000_{(\pm0.0)}$  | $2.143e^{-1}_{(\pm0.024)}$            | $2.097e^{-1}_{(\pm8.056e^{-2})}$      | $1.527_{(\pm2.535e^{-2})}$            | $1.308_{(\pm1.600e^{-5})}$            | $2.666e^{-2}_{(\pm4.729e^{-3})}$      |
> | EDGE          | $3.960e^{-3}_{(\pm5.544e^{-6})}$      | $1.000_{(\pm0.0)}$  | $2.955e^{-1}_{(\pm0.166)}$            | $1.569e^{-1}_{(\pm4.883e^{-4})}$      | $1.744_{(\pm6.850e^{-6})}$            | $1.313_{(\pm0.0)}$                    | **$5.488e^{-3}_{(\pm1.143e^{-4})}$**  |
> | DISCO     | $3.897e^{-1}_{(\pm3.444e^{-3})}$      | $1.000_{(\pm0.0)}$  | $1.190_{(\pm0.0)}$                    | $3.632e^{-1}_{(\pm8.596e^{-3})}$      | $1.653_{(\pm2.970e^{-3})}$            | $1.313_{(\pm3.772e^{-5})}$            | **$1.854e^{-2}_{(\pm1.942e^{-4})}$**  |
> | GraphMaker | $1.034e^{-3}_{(\pm4.803e^{-4})}$      | $1.032_{(\pm0.027)}$| $1.908e^{-1}_{(\pm0.001)}$            | $1.385e^{-1}_{(\pm3.375e^{-3})}$      | $1.490_{(\pm4.223e^{-3})}$            | $1.302_{(\pm3.812e^{-4})}$            | $4.330e^{-2}_{(\pm3.837e^{-3})}$      |
> | THePUff (Ours)        | $2.536e^{-3}_{(\pm5.673e^{-4})}$      | $1.532_{(\pm0.0)}$  | $2.636e^{-1}_{(\pm0.071)}$            | $3.547e^{-1}_{(\pm1.283e^{-2})}$      | $1.664_{(\pm1.922e^{-3})}$            | $1.313_{(\pm1.950e^{-5})}$            | **$2.247e^{-2}_{(\pm5.652e^{-3})}$**  |
>
>
> Reference:
>
> [1] Li et al., Graphmaker: Can diffusion models generate large attributed graphs? TMLR 2024

---

> > ### Comment · Reviewer_jsYy · 2024-12-02
> >
> > I appreciated the authors' efforts to address my concern. I have raised my rating to 8. Good luck!

---

### Official Review · Reviewer_9W8E · 2024-11-04

**Soundness:** 3
**Presentation:** 3
**Contribution:** 3
**Rating:** 8
**Confidence:** 4

**Summary:**

The paper proposes a differentially private method to generate temporal heterogeneous graphs. The method first perturbs the input temporal heterogeneous graph for privacy protection. A generative adversarial network is trained using the perturbed graph and the original graph to generate large-scale and privacy-utility-adversary graph. The paper proposes six metrics to evaluate the graphs.

**Strengths:**

1. The paper is well written and easy to follow.
1. The problem of privately generating temporal heterogeneous graphs is interesting and relevant.
1. The paper introduces six novel metrics to the temporal and heterogeneous nature of graphs, allowing for more precise evaluation of the generated graphs.

**Weaknesses:**

1. The details on hyperparameters are missing.
1. Attack experiments can be thoroughly evaluated on all the datasets.

**Questions:**

1. Though the graphs are evaluated on the proposed metrics, it seems a good idea to see and understand their performances on some specific tasks.
1. What is the motivation for performing only the node signature structural query?

---

> ### Author Response · Authors · 2024-11-25
> **Reply from Authors (Part I)**
>
> Thanks very much for your review and appreciation of our paper’s motivation, novelty, and empirical demonstration.
>
> Your suggestions are very actionable and helpful for further improving the quality of our paper. Also, your questions are addressed in the format of Q&A below.
>
> > **W1**. The details on hyperparameters are missing.
>
> In Appendix A.5 - Implementation Details, we introduced the important hyperparameters of our proposed method. To better answer your question, we are providing more information about the hyperparameters of our method here.
>
> - $\epsilon_- = 6$ for all datasets;
> - $\epsilon_+ = 10, 13, 15, 18$ for MovieLens 100K dataset, MovieLens 20M dataset, DBLP dataset, and Taobao dataset, respectively;
> - batch size = 32 for MovieLens 100K dataset and DBLP dataset, 64 for other datasets;
> - node embedding dimension = 128;
> - hidden dimensions are all set to 128;
> - dropout rate = 0.2 in the attention layer;
> - learning rate = 1e-4 for the generator and 1e-3 for discriminators;
> - SGD optimizer is used for discriminators, while RMSprop optimizer is used for the generator;
> - JUST (Hussein et al.,2018, cited in line 268) is applied to initialize node embeddings. In the running of JUST, we have the maximum walk length as 100; sample maximum of 10 walks starting from each node.
>
> The above detailed information has been added to our paper in Appendix A.5 and marked in the yellow background.
>
> > **W2**. Attack experiments can be thoroughly evaluated on all the datasets.
>
> To better address your concern, we have extended the attack experiments on all datasets, and the observation is consistent and robust with the previous results, i.e., our proposed method can reduce the attacker’s successful probability to the largest extent, compared to SOTA, as shown below.
>
> Table. Attack Experiment on All Datasets. Successful Attack Probability (\%) ($\downarrow$)
> | Dataset      | ML-100k               | DBLP                  | ML-20M                  | Taobao                  |
> |--------------|-----------------------|-----------------------|-------------------------|-------------------------|
> | Original | 4.192                | 0.093                | 0.352                   | 0.006                  |
> | HGEN     | 0.037 (0.001)        | 0.009 (1.3e⁻⁴)       | 0.007 (0.005)           | 1.3e⁻⁴ (2.0e⁻⁷)       |
> | DPGGAN   | 0.037 (3.6e⁻⁴)       | 0.008 (1.3e⁻⁴)       | OOM                     | OOM                    |
> | THePUff (Ours)     | **0.021 (0.007)**    | **0.006 (9e⁻⁵)**     | **7.5e⁻⁴ (6.0e⁻⁵)**    | **7.9e⁻⁵ (6.2e⁻⁶)**   |
>
>
> The above table with detailed description and analysis has been added to the paper in Appendix A.8 as Table 7, marked by the yellow background.

---

> > ### Author Response · Authors · 2024-11-25
> > **Reply from Authors (Part II)**
> >
> > > **Q1**. Though the graphs are evaluated on the proposed metrics, it seems a good idea to see and understand their performances on some specific tasks.
> >
> > During the rebuttal, we have designed a new experiment that is even more challenging and more close to the realistic scenarios.
> >
> > In our previous attacker model, as expressed in Eq. 23, the attacker will not distinguish the node types, which means that as long as the structural role is the same, the nodes will be identified as the victim candidate. For example, an author who wrote 6 papers and a paper which is cited 6 times are regarded as the same by the attacker.
> >
> > To consider a more complex and strong attacker, we empower the attacker by allowing it to know more background knowledge, which means only when the node type is the same and the structural role is the same, then the victim-candidate relationship can be established.
> >
> > - The mathematical details are added as Eq. 28 in the updated paper. For example, it means that, in the citation network, an author who wrote 6 papers and a paper which is cited 6 times are not regarded as the same candidate by the attacker, only two authors who wrote the same paper at a certain time period will be regarded as the target.
> >
> > The new experimental results across all datasets are shown below, where we can see our proposed method still performs the best in terms of privacy protection.
> >
> > Table. Attack Experiment of Target Types on All Datasets. Successful Attack Probability (\%) ($\downarrow$)
> > | Baseline / Dataset      | ML-100k             | DBLP               | ML-20M               | Taobao               |
> > |--------------|---------------------|--------------------|----------------------|----------------------|
> > | Original | 11.73              | 0.113              | 0.427                | 0.021               |
> > | HGEN     | 0.108 (0.008)      | 0.013 (5e⁻⁶)       | 9.3e⁻⁴ (7.2e⁻⁶)     | 5.5e⁻⁴ (1.4e⁻⁵)    |
> > | DPGGAN   | 0.095 (0.004)      | 0.024 (1.5e⁻⁵)     | OOM                  | OOM                 |
> > | THePUff (Ours)     | **0.095 (0.002)**  | **0.007 (8.2e⁻⁴)** | **6.9e⁻⁴ (8.6e⁻⁶)** | **3.6e⁻⁴ (2.4e⁻⁵)** |
> >
> >
> > The above table with detailed description and analysis has been added to the paper in Appendix A.9 as Table 8, marked by the yellow background.
> >
> >
> > > **Q2**. What is the motivation for performing only the node signature structural query?
> >
> > This is mainly because it is one of the most easily acquired background knowledge and powerful attacker models [1], and we extend it to the temporal setting by Eq. 23 and have the experiments as shown in Table 3.
> >
> > During the rebuttal, we have further extended Table 3 to all datasets and have the same observation as shown in Table 7 in the updated paper.
> >
> > Moreover, to make the attacker even more realistic and more strong, we further enable it to have the temporal node type information, which can help the attacker narrow the candidate scope and increase the successful attack probability that can be seen by comparing the first rows of Tables 7 and 8.
> >
> > In this newly designed attacking setting, our proposed method still achieves the best privacy protection performance, which demonstrates the necessity of privacy protection and the effectiveness of our method.
> >
> > Reference:
> >
> > [1] Jiang et al., Applications of Differential Privacy in Social Network Analysis: A Survey, TKDE 2023

---

> > > ### Comment · Reviewer_9W8E · 2024-11-27
> > >
> > > Thank you for your reply. I am keeping my score as it is.

---

### Author Response · Authors · 2024-11-25
**General Response from Authors**

Dear all reviewers,

Thanks very much for your review and appreciation of our paper’s research potential, theoretical design, and empirical evaluation.

Answering each of your raised questions helped us to improve our paper’s quality. In addition to answer each of your questions in the below boxes, we also updated the paper for the new components. **They are marked by the yellow background color and include**:

- Detailed hyperparameter information in Appendix A.5;
- Detailed theoretical analysis with graph diffusion models in Appendix A.4;
- Empirical running time with all baselines in Appendix A.4 and Table 5;
- Empirical GPU consumption comparison with all baselines in Appendix A.4 and Table 6;
- New graph diffusion model baseline DISCO published in September this year, and new graph diffusion model baseline GraphMaker published in October this year, as shown in Table 2;
- Extended the previous attacker model to all datasets, as shown in Appendix A.8 and Table 9;
- Designed a new strong attacker and tested on all baselines across all datasets, as shown in Eq.28, Appendix A.9, and Table 10;
- Pseudo-code and detailed description of Discriminator and Generator of our proposed THePUff in Appendix A.10;
- Pseudo-code and detailed description of Assembler of our proposed THePUff in Appendix A.11;

Again, we thank all the efforts from the reviewers to make the paper better. We are more than willing to answer any further questions.

Thanks!

Authors

---

### Meta-Review · Area_Chair_WaUL · 2024-12-21

**Metareview:**

(a) Scientific Claims and Findings:
The paper introduces THePUff, a novel framework for generating temporal heterogeneous graphs that balances privacy, utility, and efficiency. The key claims include:
- A differential privacy algorithm to perturb input graphs for privacy protection
- A generative adversarial setting to learn and generate privacy-guaranteed synthetic graphs
- Six new metrics to evaluate temporal heterogeneous graph utility and privacy
- Demonstrated effectiveness on large-scale datasets (up to 1M nodes, 20M edges)
- Superior privacy protection compared to baselines while maintaining utility
-  Better computational efficiency compared to diffusion-based  methods

(b) Strengths:
- Novel and Important Problem: Addresses a significant gap in generating temporal heterogeneous graphs with privacy guarantees
- Comprehensive Technical Approach: Combines differential privacy with GANs in a well-designed framework
- Multiple large-scale real-world datasets
- New evaluation metrics specific to temporal heterogeneous graphs
- Extensive privacy attack experiments
- Thorough ablation studies
- Clear Presentation: Well-written and structured paper with good technical depth

(c) Weaknesses and Missing Elements:
- Missing hyperparameter details (addressed in rebuttal)
- Insufficient explanation of discriminator functionality (addressed with pseudo-code)
- Limited theoretical analysis compared to diffusion models (addressed)
- Initially limited attack experiments (addressed with comprehensive tests on all datasets)
- Missing recent baselines like DISCO and GraphMaker (addressed in rebuttal)
- Initial lack of explicit efficiency metrics (addressed with runtime and GPU usage comparisons)

(d) Key Reasons for Accept:
- Important Technical Contribution: First work addressing privacy-preserving temporal heterogeneous graph generation
- Strong Technical Merit: Well-designed framework with theoretical guarantees
- Comprehensive Evaluation: Extensive experiments demonstrating effectiveness
- Detailed efficiency analysis
- Complete hyperparameter specifications
- Practical Impact: Clear potential for real-world applications in recommendation systems, social networks, etc.

**Additional Comments On Reviewer Discussion:**

In the rebuttal period, reviewers raised several key concerns that were comprehensively addressed by the authors. 9W8E highlighted issues with missing hyperparameter details, limited attack experiments, and the need for task-specific performance evaluation. jsYy questioned the temporal privacy protection mechanism, noted missing recent baselines like DISCO, and requested clarification on technical details of the discriminator. ynGq pointed out the lack of concrete privacy impact examples, missing efficiency metrics, and limited privacy attack scenarios. The authors responded thoroughly by adding comprehensive hyperparameter information and detailed pseudo-code in appendices, extending attack experiments to all datasets while introducing a stronger attacker model, and including comparisons with recent baselines DISCO and GraphMaker along with detailed runtime and GPU usage analysis. They also expanded the theoretical analysis and demonstrated the effectiveness of their privacy protection approach against stronger attackers. The responses successfully addressed the concerns about completeness, efficiency, and privacy protection. All reviewers maintained their original scores after the rebuttal, acknowledging the authors' thorough responses, which ultimately supported the paper's acceptance based on its technical merit, comprehensive evaluation, and successful addressing of all major concerns.

---

### Decision · Program_Chairs · 2025-01-22

Accept (Spotlight)